# Validation of lake surface state in the HIRLAM v.7.4 NWP model against *in-situ* measurements in Finland

**Laura Rontu[1], Kalle Eerola[1], and Matti Horttanainen[1]**

[1]Finnish Meteorological Institute, P.O. Box 503, 00101 Helsinki, Finland

**Correspondence:** laura.rontu@fmi.fi

**Abstract.** High Resolution Limited Area Model (HIRLAM), used for the operational numerical weather prediction in the Finnish Meteorological Institute (FMI), includes prognostic treatment of lake surface state since 2012. Forecast is based on the Freshwater Lake (FLake) model integrated into HIRLAM. Additionally, an independent objective analysis of lake surface water temperature (LSWT) combines the short forecast of FLake to observations from the Finnish Environment Institute (SYKE). The resulting description of lake surface state - forecast FLake variables and analysed LSWT - was compared to SYKE observations of lake water temperature, freeze-up and break-up dates as well as the ice thickness and snow depth for 2012-2018 over 45 lakes in Finland. During the ice-free period, the predicted LSWT corresponded to the observations with a slight overestimation, with a systematic error of + 0.91 K. The colder temperatures were underrepresented and the maximum temperatures were too high. The objective analysis of LSWT was able to reduce the bias to + 0.35 K. The predicted freeze-up dates corresponded well the observed dates, mostly within the accuracy of a week. The forecast break-up dates were far too early, typically several weeks ahead of the observed dates. The growth of ice thickness after freeze-up was generally overestimated. However, practically no predicted snow appeared on lake ice. The absence of snow, presumably be due to an incorrect security coefficient value, is suggested to be also the main reason of the inaccurate simulation of the lake ice melting in spring.

## 1 Introduction

Lakes influence the energy exchange between the surface and the atmosphere, the dynamics of the atmospheric boundary layer and the near-surface weather. This is important for weather forecasting over the areas where lakes, especially those with a large yearly variation of the water temperature, freezing in autumn and melting in spring, cover a significant area of the surface (Kheyrollah Pour et al., 2017; Laird et al., 2003 and references therein). Description of the lake surface state influences the numerical weather prediction (NWP) results, in particular in the models whose resolution is high enough to account for even the smaller lakes (Eerola et al., 2014 and references therein). Especially, the existence of ice can be important for the numerical forecast (Eerola et al., 2014; Cordeira and Laird, 2008).

In the Finnish Meteorological Institute (FMI), the High Resolution Limited Area Model HIRLAM (Undén et al., 2002; Eerola, 2013) has been applied since 1990 for the numerical short-range weather forecast. In the beginning, the monthly climatological water surface temperature for both sea (Sea Surface Temperature SST) and lakes (Lake Surface Water Temperature LSWT) was used. Since 2012, HIRLAM includes a prognostic lake temperature parameterization based on the Freshwater Lake Model (FLake, Mironov et al., 2010). An independent objective analysis of observed LSWT (Kheyrollah Pour et al., 2017 and references therein) was implemented in 2011. The fractional ice cover (lake ice concentration in each gridsquare of the model) is diagnosed from the analysed LSWT.

FLake was designed to be used as a parametrization scheme for the forecast of the lake surface state in NWP and climate models. It allows to predict the lake surface state in interaction with the atmospheric processes treated by the NWP model. The radiative and turbulent fluxes as well as the predicted snow precipitation from the atmospheric model

are combined with FLake processes at each time-step of the model integration in the model grid, where the fraction and depth of lakes are prescribed.

FLake has been implemented into the other main European NWP and regional climate models, first into COSMO (Mironov et al., 2010) then into ECMWF (Balsamo et al., 2012), Unified Model (Rooney and Bornemann, 2013), SUR-FEX surface modelling framework (Masson et al., 2013), regional climate models RCA (Samuelsson et al., 2010), HCLIM (Lindstedt et al., 2015) and REMO (Pietikäinen et al., 2018), among others. Description of lake surface state and its influence in the numerical weather and climate prediction has been validated in various ways. Results of case studies, e.g. Eerola et al. (2014) and shorter-period NWP experiments, e.g. Eerola et al. (2010); Rontu et al. (2012); Kheyrollah Pour et al. (2014); Kheyrollah Pour et al. (2017) as well as climate model results, e.g. Samuelsson et al. (2010); Pietikäinen et al. (2018), have been compared with remote-sensing satellite data and *in-situ* lake temperature and ice measurements as well as validated against the standard weather observations. In general, improvement of the scores has been seen over regions where lakes occupy a significant area. However, specific features of each of the host models influence the results of the coupled atmosphere-lake system as FLake is quite sensitive to the forcing by the atmospheric model.

The aim of the present study is to validate the lake surface state forecast by the operational HIRLAM NWP model using the *in-situ* LSWT measurements, lake ice freeze-up and break-up dates and measurements of ice and snow thickness by the Finnish Environment Institute (Suomen Ympäristökeskus = SYKE). For this purpose, HIRLAM analyses and forecasts archived by FMI were compared with the observations by SYKE over the lakes of Finland from spring 2012 to summer 2018. To our knowledge, this is the longest available detailed dataset that allows to evaluate how well the lake surface state is simulated by an operational NWP model that applies FLake parametrizations.

## 2 Lake surface state in HIRLAM

FLake was implemented in the HIRLAM forecasting system in 2012 (Kourzeneva et al., 2008; Eerola et al., 2010). The model utilizes external datasets on the lake depth (Kourzeneva et al., 2012a; Choulga et al., 2014) and the lake climatology (Kourzeneva et al., 2012b). The latter is only needed in order to provide initial values of FLake prognostic variables in the very first forecast (so-called cold start). The use of real-time *in-situ* LSWT observations by SYKE for 27 Finnish lakes was introduced in 2011 into the operational LSWT analysis in HIRLAM (Eerola et al., 2010; Rontu et al., 2012). In the current operational HIRLAM of FMI, FLake provides the background for the optimal interpolation analysis (OI, based on Gandin, 1965) of LSWT. How-

ever, the prognostic FLake variables are not corrected using the analysed LSWT. This would require more advanced data assimilation methods based on e.g. the extended Kalman filter (Kourzeneva, 2014).

### 2.1 Freshwater lake model in HIRLAM

FLake is a bulk model capable of predicting the vertical temperature structure and mixing conditions in lakes of various depths on time-scales from hours to years (Mironov et al., 2010). The model is based on two-layer parametric representation of the evolving temperature profile in the water and on the integral budgets of energy for the layers in question. Bottom sediments and the thermodynamics of the ice and snow on ice layers are treated separately. FLake depends on prescribed lake depth information. The prognostic and diagnostic variables of HIRLAM FLake together with the analysed lake surface variables in HIRLAM are listed in the Appendix (Table A1).

At each time step during the HIRLAM forecast, FLake is driven by the atmospheric radiative and turbulent fluxes as well as the predicted snowfall, provided by the physical parameterisations in HIRLAM. This couples the atmospheric variables over lakes with the lake surface properties as provided by FLake parametrization. Most importantly, FLake provides HIRLAM with the evolving lake surface (water, ice, snow) temperature and radiative properties, that influence the HIRLAM forecast of the grid-average near-surface temperatures, humidity and wind.

Implementation of FLake model as a parametrization scheme in HIRLAM was based on the experiments described by Rontu et al. (2012). Compared to the reference version of FLake (Mironov et al., 2010), minor modifications were introduced, namely, use of constant snow density = 300 $kgm^{-3}$, molecular heat conductivity = 1 $Jm^{-1}s^{-1}K^{-1}$, constant albedos of dry snow = 0.75 and ice = 0.5. Bottom sediment calculations were excluded. Global lake depth database (GLDB v.2, Choulga et al., 2014) was used for derivation of mean lake depth in each gridsquare. Fraction of lake was taken from HIRLAM physiography database, where it originates from GLCC (Loveland et al., 2000).

Lake surface temperature is diagnosed from the mixed layer temperature for the unfrozen lake gridpoints and from the ice or snow-on-ice temperature for the frozen points. In FLake, ice starts to grow from an assumed value of one millimeter when temperature reaches the freezing point. The whole lake tile in a gridsquare is considered by FLake either frozen or unfrozen. Snow on ice is accumulated from the model's snowfall at each time step during the numerical integration.

### 2.2 Objective analysis of LSWT observations

A comprehensive description of the optimal interpolation (OI) of the LSWT observations in HIRLAM is given by

Kheyrollah Pour et al., 2017. Shortly, LSWT analysis is obtained by correcting the FLake forecast at each gridpoint by using the weighted average of the deviations of observations from their background values. Prescribed statistical informa-[5]tion about the observation and background error variance as well as the distance-dependent autocorrelation between the locations (observations and gridpoints) are applied. The real-time observations entering the HIRLAM surface analysis system are subject to quality control in two phases. First,[10]the observations are compared to the background, provided by the FLake short forecast. Second, optimal interpolation is done at each observation location, using the neighbouring observations only (excluding the current observation) and comparing the result to the observed value at the station.

[15]A specific feature of the lake surface temperature OI is that the interpolation is performed not only within the (large) lakes but also across the lakes: within a statistically predefined radius, the observations affect all gridpoints containing a fraction of lake. This ensures that the analysed LSWT[20]on lakes without own observations may also be influenced by observations from neighbouring lakes, not only by the first guess provided by FLake forecast.

The relations between the OI analysis and the prognostic FLake in HIRLAM are schematically illustrated in Figure 1.[25]Within the present HIRLAM setup, the background for the analysis is provided by the short (6-hour) FLake forecast but the next forecast is not initialized from the analysis. Instead, FLake continues running from the previous forecast, driven by the atmospheric state given by HIRLAM at each time step.[30]This means that FLake does not benefit from the result of OI analysis but the analysis remains as an extra diagnostic field, to some extent independent of the LSWT forecast. However, FLake background has a large influence in the analysis, especially over distant lakes where neighbouring observations[35]are not available. The diagnostic LSWT analysis, available at every gridpoint of HIRLAM, might be useful e.g. for hydrological, agricultural or road weather applications.

Missing LSWT observations in spring and early winter are interpreted to represent presence of ice and given a flag value[40]of -1.2°C. If, however, the results of the statistical LSWT model (Elo, 2007), provided by SYKE along with the real-time observations, indicate unfrozen conditions, the observations are considered missing. This prevents appearance of ice in summer if observations are missing but leads to a misin-[45]terpretation of data in spring if the SYKE model indicates too early melting. In the analysis, fraction of ice is diagnosed from the LSWT field in a simple way. The lake surface within a gridsquare is assumed fully ice-covered when LSWT falls below -0.5°C and fully ice-free when LSWT is[50]above 0°C. Between these temperature thresholds, the fraction of ice changes linearly (Kheyrollah Pour et al., 2014).

The HIRLAM surface data assimilation system produces comprehensive feedback information from every analysis-forecast cycle. The feedback consists of the observed value[55]and its deviations from the background and from the final

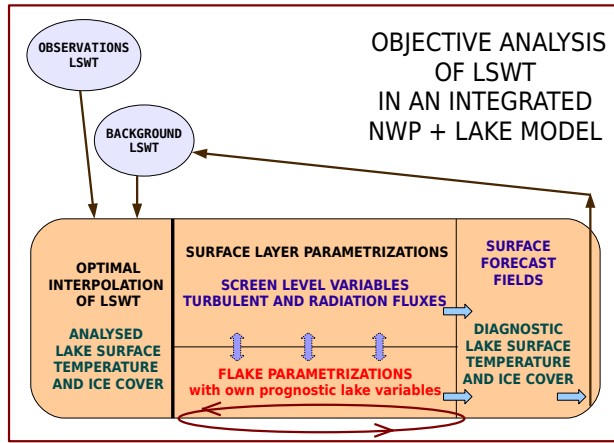

**Figure 1.** Coexistence of the independent objective analysis of the observed LSWT and prognostic FLake parametrizations in HIRLAM. The thin arrows are related to data flow between HIRLAM analysis-forecast cycles while the thick arrows describe processes within each cycle.

analysis at the observation point. Bilinear interpolation of the analysed and forecast values is done to the observation location from the nearest gridpoints that contain a fraction of lake. In addition, information about the quality check and usage of observations is provided. Fractions of land and lake[60]in the model grid as well as the weights, which were used to interpolate gridpoint values to the observation location, are given. This information is the basis of the present study (see sections 3.3 and 4).

## 3 Model-observation intercomparison 2012-2018 [65]

In this intercomparison we validated HIRLAM results against observations about the lake surface state. The impact of FLake parametrizations to the weather forecast by HIRLAM was not considered. This is because no non-FLake weather forecasts exist for comparison with the operational[70]forecasts during the validation period.

Throughout the following text, the analysed LSWT refers to the result of OI analysis, where FLake forecast has been used as background (Section 2.2) while the forecast LSWT refers to the value diagnosed from the mixed layer water tem-[75]perature predicted by FLake (Section 2.1). Observed LSWT refers to the measured by SYKE lake water temperature (Section 3.2).

### 3.1 FMI operational HIRLAM

FMI operational HIRLAM is based on the last reference[80]version (v.7.4), implemented in spring 2012. (Eerola, 2013 and references therein). FLake was introduced into this version. After that further development of HIRLAM model has

been stopped. Thus, during the years of the present comparison, the FMI operational HIRLAM system remains unmodified, which offers a clean time series of data for the model-observation intercomparison. The general properties of the system are summarised in Table 1.

## 3.2 SYKE lake observations

In this study we used three different types of SYKE lake observations: LSWT, freeze-up and break-up dates and ice thickness and snow depth on lake ice. In total, observations on 45 lakes listed in Appendix (Table A2) were included as detailed in the following. The lake depths and surface areas given in Table A2 are based on the updated lake list of GLDB v.3 (Margarita Choulga, personal communication).

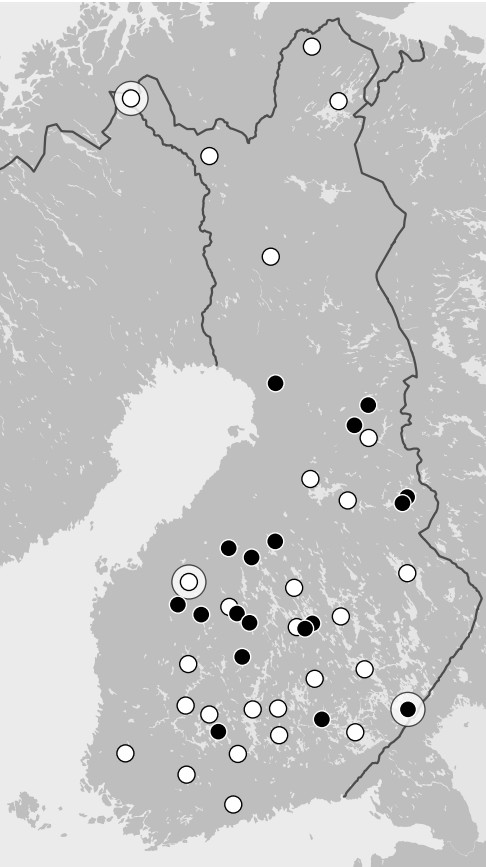

**Figure 2.** Map of SYKE observation points used in this study: lakes with both lake surface water temperature (LSWT) and lake ice date (LID) observations (white), lakes where only LID is available (black). On Lakes Lappajärvi, Kilpisjärvi and Simpelejärvi also ice thickness and snow depth measurements were used (Section 4.3), they are surrounded with a large white circle. List of the lakes with coordinates is given in Appendix A2.

### 3.2.1 Lake temperature measurements

Regular *in-situ* lake water temperature measurements are performed by SYKE. Currently SYKE operates 34 regular lake and river water temperature measurement sites in Finland. The temperature of the lake water is measured every morning at 8.00 AM local time, close to shore, at 20 cm below the water surface. The measurements are recorded either automatically or manually and are performed only during the ice-free season (Korhonen, 2019). Further, we will for simplicity denote also these data as LSWT observations although they do not represent exactly the same surface water temperature (skin temperature, radiative temperature) that could be estimated by satellite measurements. These data are available in the SYKE open data archive (SYKE, 2018). Measurements from 27 of these 34 lakes (Figure 2, white dots) were selected for use in the FMI operational HIRLAM in 2011, and the list has been kept unmodified since that. The set of 27 daily observations, quality-controlled by HIRLAM, were obtained from the analysis feedback files and used in all comparisons reported in this study.

### 3.2.2 Freeze-up and break-up dates

Regular visual observations of freeze-up and break-up of lakes have been recorded in Finland for centuries, the longest time series starting in the middle of the 19th century (Korhonen, 2019). Presently, dates of freeze-up and break-up are available from SYKE (2018) on 123 lakes, but the time series for many lakes are discontinuous. Further, we will denote the break-up and freeze-up dates together by "lake ice dates" (LID). LID observations aim at representing conditions on entire lakes. For both freeze-up and break-up the dates are available in two categories (terminology from Korhonen, 2019): "freeze-up of the lake within sight" (code 29 by SYKE) and "freeze-up of the whole lake" (code 30). For break-up the dates are defined as "no ice within sight" (code 28) and "thaw areas out of the shore" (code 27). LID observations by SYKE are made independently of their LSWT measurements and possibly from different locations on the same lakes. Therefore the LSWT measurements may be started later than the date of reported lake ice break-up or end earlier than the reported freeze-up date.

LID from the 27 lakes whose LSWT measurements are used in HIRLAM were available and selected for this study. In addition, 18 lakes with only LID available (Figure 2, black dots) were chosen for comparison with HIRLAM LID.

### 3.2.3 Ice thickness and snow depth on lakes

In the period 2012-2018 SYKE recorded the lake ice thickness and snow depth on around 50 locations in Finland. (Archived historical data are available in total from 160 measurement sites). The manual measurements are done three times a month during the ice season. Thickness of ice and

**Table 1.** FMI operational HIRLAM

| | |
|---|---|
| Domain | From Atlantic to Ural, from North Africa beyond North Pole |
| Model horizontal / vertical resolution | 7 km / 65 levels |
| HIRLAM version | 7.4 |
| Model dynamics | Hydrostatic, semi-Lagrangian, grid-point |
| Atmospheric physical parametrizations | Savijärvi radiation, CBR turbulence, |
| | Rasch-Kristiansson cloud microphysics + Kain-Fritsch convection |
| Surface physical parametrizations | ISBA-newsnow for surface, FLake for lakes |
| Data assimilation | Default atmospheric (4DVAR) and surface (OI) analysis |
| Lateral boundaries | ECMWF forecast |
| Forecast | Up to +54 h initiated every 6h (00, 06, 12, 18 UTC) |

snow depth on ice are measured by drilling holes through snow and ice layers along chosen tracks, normally at least 50 m from the coast (Korhonen, 2019). The locations may differ from those of the LSWT measurement or LID observation over the same lakes.

### 3.3 Validation of HIRLAM lake surface state

#### 3.3.1 Lake surface water temperature

LSWT by HIRLAM, resulting from the objective analysis or diagnosed from the forecast, was compared with the observed LSWT by SYKE using data extracted from the analysis feedback files (Section 2.2) at the observation locations on 06 UTC every day, excluding the winter periods 1 December - 31 March. The observations (ob) at 27 SYKE stations were assumed to represent the true value, while the analysis (an) is the result of OI that combines the background forecast (fc) with the observations. Time-series, maps and statistical scores, to be presented in Section 4.1, were derived from these.

#### 3.3.2 Lake ice conditions

For this study, the observed LID, ice and snow thickness observations were obtained from SYKE open data base, relying on their quality control. The analysed LSWT as well as the predicted ice thickness and snow depth were picked afterwards from the HIRLAM archive for a single gridpoint nearest to each of the 45 observation locations (not interpolated as in the analysis feedback file that was used for the LSWT comparison). It was assumed that the gridpoint value nearest to the location of the LSWT observation represents the ice conditions over the chosen lake.

LID given by HIRLAM were defined in two independent ways: from the analysed LSWT and from the forecast lake ice thickness. Note that the ice thickness and snow depth on ice are not analysed variables in HIRLAM. In autumn a lake can freeze and melt several times before final freeze-up. The last date when the forecast ice thickness crossed a critical value of 1 mm or the analysed LSWT fell below freezing point was selected as the date of freeze-up. In the same way,

the last date when the forecast ice thickness fell below the critical value of 1 mm or the analysed LSWT value crossed the freezing point was selected as break-up date. To decrease the effect of oscillation of the gridpoint values between the HIRLAM forecast-analysis cycles, the mean of the four daily ice thickness forecasts or analysed LSWT values was used.

LID by HIRLAM were compared to the observed dates during 2012-2018. In this comparison we included data also during the winter period. The category 29 observations ("freeze-up of the lake within sight", see Section 3.2.2) were used. In this category the time series were the most complete at the selected stations. For the same reason, the break-up observations of category 28 ("no ice within sight") were used for comparison. Furthermore, using a single gridpoint value for the calculation of LID also seems to correspond best the observation definition based on what is visible from the observation site. The statistics were calculated as fc - ob and an - ob. Hence, positive values mean that break-up or freeze-up takes place too late in the model as compared to the observations.

Lake ice thickness and snow depth measurements from lakes Lappajärvi, Kilpisjärvi and Simpelejärvi were utilised as additional data for validation of predicted by HIRLAM ice thickness and snow depth (Section 4.3). These lakes, representing the western, northern and south-eastern Finland, were selected for illustration based on the best data availability during the study years. They are also sufficiently large in order to fit well the HIRLAM grid.

## 4 Results

### 4.1 Analysed and forecast LSWT at observation points

Figure 3 shows the frequency distribution of LSWT according to FLake forecast and SYKE observations. It is evident that the amount of data in the class of temperatures which represents frozen conditions (LSWT flag value 272 K) was underestimated by the forecast (Figure 3a). When subzero LSWT values were excluded from the comparison (Figure 3b), underestimation in the colder temperature classes and overestimation in the warmer classes still remains.

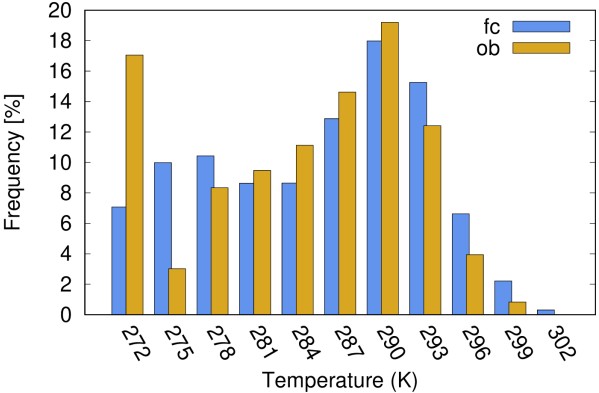

(a) with all temperatures (also frozen conditions) included

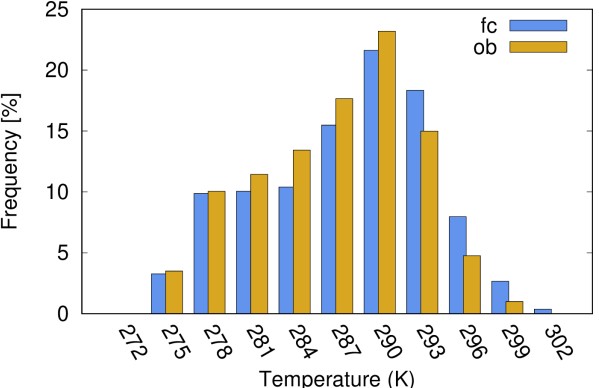

(b) only open water temperatures included

**Figure 3.** Frequency of observed (ob, yellow) and forecast (fc, blue) LSWT over all 27 SYKE lakes 2012-2018. x-axis: LSWT, unit K, classified in three-degree intervals from 270.1 to 303.1 K, y-axis: frequency, unit %.

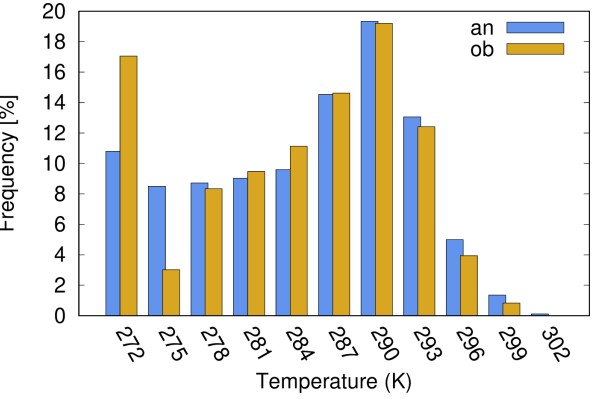

(a) with all temperatures (also frozen conditions) included

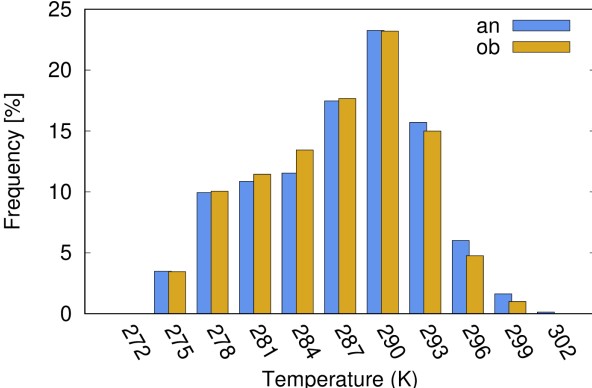

(b) only open water temperatures included

**Figure 4.** As for Figure 3 but for observed and analysed (an) LSWT.

LSWT analysis (Figure 4) improved the distribution somewhat but the basic features remain. This is due to the dominance of FLake forecast via the background of the analysis. In Section 4.3, we will show time-series illustrating the physics behind these LSWT statistics.

Table 2 confirms the warm bias by FLake in the unfrozen conditions. Similar results were obtained for all stations together and also for our example lakes Lappajärvi and Kilpisjärvi, to be discussed in detail in Section 4.3. There were three lakes with negative LSWT bias according to FLake forecast, namely the large lakes Saimaa and Päijänne and the smaller Ala-Rieveli. After the correction by objective analysis, a small positive bias converted to negative over 6 additional lakes, among them the large lakes Lappajärvi in the west and Inari in the north. The mean absolute error decreased from forecast to analysis on every lake.

In the frequency distributions, the warm temperatures were evidently related to summer. For FLake, the overestimation of maximum temperatures, especially in shallow lakes, is a known feature (e.g. Kourzeneva 2014). It is related to the difficulty of forecasting the mixed layer thermodynamics under strong solar heating and possibly to the effect of the direct radiative heating of the bottom sediments. Cold and subzero temperatures occurred in spring and autumn. In a few large lakes like Saimaa, Haukivesi, Pielinen, LSWT tended to be slightly underestimated in autumn both according to the FLake and the analysis (not shown). The cold left-hand side columns in the frequency distributions (Figures 3a and 4a) are mainly related to spring, when HIRLAM tended to melt the lakes significantly too early (Sections 4.2 and 4.3).

There are problems, especially in the analysed LSWT, over (small) lakes of irregular form that fit poorly the HIRLAM grid and where the measurements may represent more the local than the mean or typical conditions over the lake. These are the only ones where an underestimation of summer LSWT was seen. Cases occurred where FLake results differ so much from the observations that the HIRLAM quality control against background values rejected the observations, forcing also the analysis to follow the incorrect forecast (not shown).

**Table 2.** Statistical scores for LSWT at all stations and at two selected stations

| station unit | fc or an | mean ob K | bias K | mae K | stde K | N |
|---|---|---|---|---|---|---|
| ALL | fc | 286.3 | 0.91 | 1.94 | 2.34 | 30877 |
| | an | 286.3 | 0.35 | 1.32 | 1.72 | 30861 |
| Lappajärvi | fc | 286.9 | 0.33 | 1.23 | 1.62 | 1243 |
| | an | 286.9 | -0.65 | 1.06 | 1.10 | 1243 |
| Kilpisjärvi | fc | 281.7 | 1.82 | 2.13 | 2.15 | 780 |
| | an | 281.7 | 1.10 | 1.42 | 1.51 | 780 |

Statistics over days when both forecast/analysis and observation indicate unfrozen conditions. bias = systematic difference fc/an - ob, mae = mean absolute error, stde = standard deviation of the error, N = number of days (06 UTC comparison, no ice).

**Table 3.** Statistical measures of the error of freeze-up and break-up date

| unit | | bias days | stde days | max days | min days | N |
|---|---|---|---|---|---|---|
| Freeze-up | LSWT an | -3.5 | 17.9 | 64 | -52 | 233 |
| | IceD fc | -0.3 | 17.8 | 67 | -41 | 233 |
| Break-up | LSWT an | -15.2 | 8.5 | 2 | -54 | 288 |
| | IceD fc | -20.5 | 9.2 | -1 | -56 | 258 |

Denotation: LSWT an - LID estimated from analysed LSWT, IceD fc - LID estimated from forecast ice thickness. bias = systematic difference fc/an - ob, stde = standard deviation of the error, max and min = maximum and minimum errors of dates during the ice seasons 2012-2018, N = number of days

## 4.2 Freeze-up and break-up dates

In this section the freeze-up and break-up dates from HIRLAM are verified against corresponding observed dates over 45 lakes (Appendix Table A2). In the following, 'LSWT an' refers to the LID estimated from analysed LSWT and 'IceD fc' to those estimated from the forecast ice thickness by FLake. The time period contains six freezing periods (from autumn 2012 to autumn 2017) and seven melting periods (from spring 2012 to spring 2018). Due to some missing data the number of freeze-up cases was 233 and break-up cases 258. The 'IceD fc' data for the first melting period in spring 2012 was missing. The overall statistics of the error in freeze-up and break-up dates are shown in Table 3. In most cases the difference in error between the dates based on forecast and analysis was small. This is natural as the first guess of the LSWT analysis is the forecast LSWT by FLake. We will discuss next the freeze-up, then the break-up dates.

The bias in the error of freeze-up dates was small according to both 'IceD fc' and 'LSWT an', -0.3 and -3.5 days, respectively. The minimum and maximum errors were large in both cases: the maximum freeze-up date occurred about two months too late, the minimum about one and a half months too early. However, as will be shown later, the largest errors mostly occurred on a few problematic lakes while in most cases the errors were reasonable.

Figure 5a shows the frequency distribution of the error of freeze-up dates. Forecast freeze-up dates occurred slightly more often in the unbiased class (error between -5 - +5 days), compared to the estimated dates from the analysis. Of all cases 48% / 40% (percentages here and in the following are given as 'IceD fc' / 'LSWT an') fell into this class. In 20% / 26% of cases the freeze-up occurred more than five days too late and only in 11% / 9% cases more than two weeks too late. In case of 'IceD fc', the class of freeze-up more than 15 days too late comprised 25 cases distributed over 15 lakes, thus mostly one or two events per lake. This suggests that the error was related more to individual years than to systematically problematic lakes. It is worth noting, that of the eight cases where the error was over 45 days, six cases were due to a single lake, Lake Kevojärvi. This lake is situated in the northernmost Finland. It is very small and narrow, with an area of 1 km$^2$, and located in a steep canyon. Therefore it is poorly represented by the HIRLAM grid (grid-square almost 50 km$^2$), and the results seem unreliable.

Concerning too early freezing, in 33% / 44% of the cases freeze-up occurred more than five days too early and in 15% / 19% more than two weeks too early. According to the forecast, these 15% (34 cases) were distributed over 19 lakes. Each of the five large lakes Pielinen, Kallavesi, Haukivesi, Päijänne and Inari occurred in this category three times while all other lakes together shared the remaining 19 cases during the six winters.

The break-up dates (Table 3) show a large negative bias, about two ('LSWT an') or three weeks ('IceD fc'), indicating that lake ice break-up was systematically forecast to occur too early. However, the standard deviation of the error was only about half of that of the error of freeze-up dates and there were no long tails in the distribution (Figure 5b). Hence the distribution is strongly skewed towards too early break-up, but much narrower than that of freeze-up (Figure 5a). The large bias was most probably due missing snow over lake ice in this HIRLAM version (see Section 5). The maximum frequency (47 %) was in the class -24 - -15 days for 'IceD fc', while in case of 'LSWT an', the maximum frequency (52 %) occurred in the class -14 - -5 days. FLake forecast 'IceD fc' suggested only three cases in the unbiased class - 4 - +5 while according to 'LSWT an' there were 12 cases in this class. Hence, the break-up dates derived from analysed LSWT corresponded the observations better than those derived from FLake ice thickness forecast.

Note that this kind of method of verifying LID compares two different types of data. The observations by SYKE are visual observations from the shore of the lake (see Section 3.2.2), while the freeze-up and break-up dates from HIRLAM are based on single-gridpoint values of LSWT or ice thickness (see Section 3.3.2). In addition, the resulting freeze-up and break-up dates from HIRLAM are somewhat sensitive to definition of the freezing and melting tresholds.

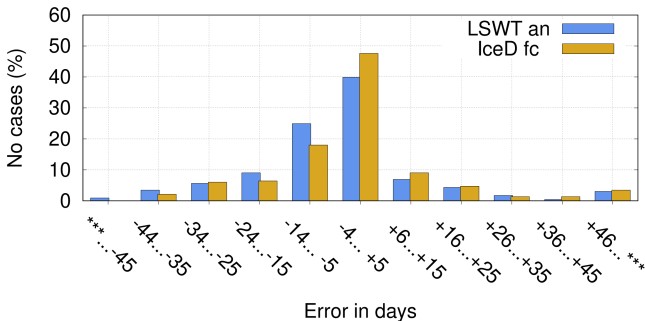

(a) error of freeze-up dates

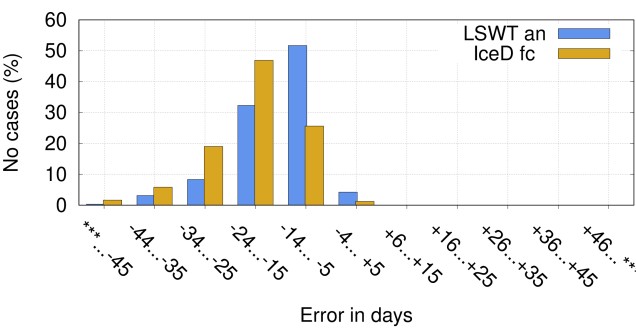

(b) error of break-up dates

**Figure 5.** Frequency distribution of the difference between analysed/forecast and observed freeze-up and break-up dates over all lakes 2012-2018. Variables used in diagnosis of ice existence: analysed LSWT crossing the freezing point (blue) and forecast ice thickness > 1 mm (yellow). Observed variable: freeze-up date by SYKE. x-axis: difference (fc-ob), unit day, y-axis: percentage of all cases.

Here we used 1 mm for the forecast ice thickness and the freezing point for the LSWT analysis as the critical values.

In conclusion, the validation statistics show that HIRLAM succeeded rather well in predicting freezing of Finnish lakes. Almost in half of the cases the error was less than $\pm$ 5 days. Some bias towards too early freeze-up can be seen both in forecast and in the analysis. Melting was more difficult. FLake predicted lake ice break-up always too early, with a mean error of over two weeks, and the analysis mostly followed it.

### 4.3   Comparisons on three lakes

In this section we present LSWT and LID time-series for two representative lakes, Kilpisjärvi in the north and Lappajärvi in the west (see the map in Figure 2). Observed and forecast ice and snow thickness are discussed, using also additional data from Lake Simpelejärvi in southeastern Finland.

Lake Kilpisjärvi is an Arctic lake at the elevation of 473 m, surrounded by fells. The lake occupies 40 % of the area of HIRLAM gridsquare covering it (the mean elevation of the

gridsquare is 614 m). The average/maximum depths of the lake are 19.5/57 m and the surface area is 37.3 km$^2$. The heat balance as well as the ice and snow conditions on Lake Kilpisjärvi have been subject to several studies (Leppäranta et al., 2012; Lei et al., 2012; Yang et al., 2013). Typically, the ice season lasts there seven months from November to May. Lake Lappajärvi is formed from a 23 km wide meteorite impact crater, which is estimated to be 76 million years old. It is Europe's largest crater lake with a surface area of 145.5 km$^2$ and an average/maximum depth of 6.9/36 m. Here the climatological ice season is shorter, typically about five months from December to April. The average/maximum depth of Lake Simpelejärvi is 8.7/34.4 m and the surface area 88.2 km$^2$. This lake is located at the border between Finland and Russia and belongs to the catchment area of Europe's largest lake, Lake Ladoga in Russia.

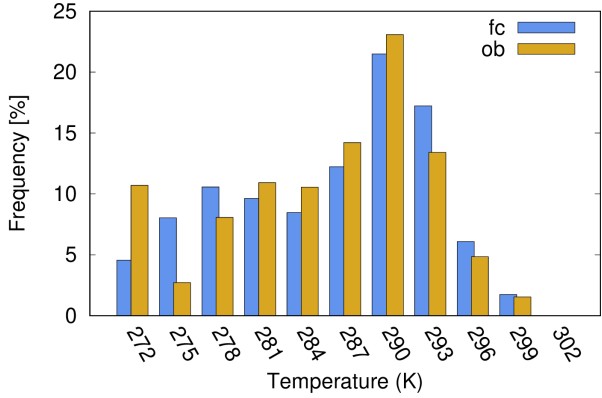

(a) forecast v.s. observation

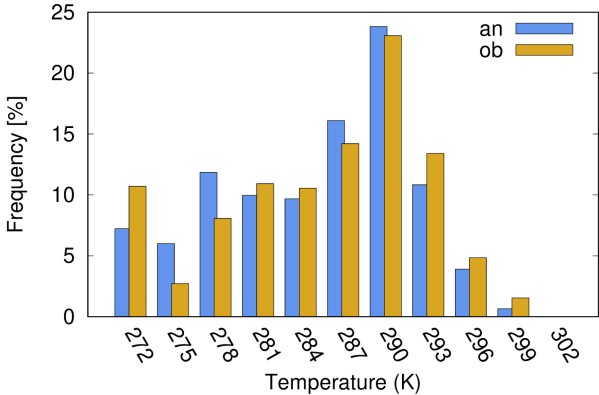

(b) analysis v.s. observation

**Figure 6.** Frequency of observed (yellow) and forecast or analysed (blue) LSWT over Lake Lappajärvi 2012-2018, all temperatures included. x-axis: LSWT, unit K, y-axis: frequency, unit %.

Figures 6 and 7 show the frequency distributions of LSWT according to forecast v.s. observation and analysis v.s. observation for Lappajärvi and Kilpisjärvi. Features similar to the results averaged over all lakes (Section 4.1, Figures 3 and 4)

are seen, i.e. underestimation of the amount of cold temperature cases and overestimation of the warmer temperatures by the forecast and analysis. On Lake Lappajärvi, only the amount of below-freezing temperatures was clearly underestimated, otherwise the distributions look quite balanced. According to the observations, on Lake Kilpisjärvi ice-covered days dominated during the periods from November to May. According to both LSWT analysis and forecast the amount of these days was clearly smaller in HIRLAM.

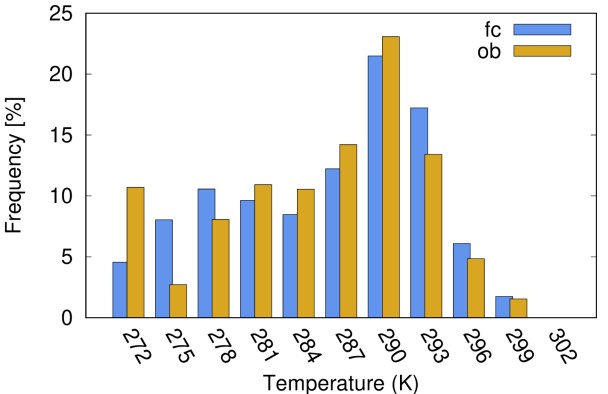

(a) forecast v.s. observation

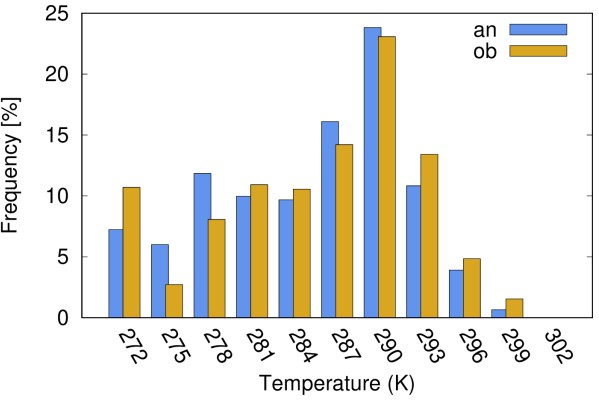

(b) analysis v.s. observation

**Figure 7.** As for Figure 6 but for Lake Kilpisjärvi.

Yearly time series of the observed, forecast and analysed LSWT, with the observed LID marked, are shown in Figures 9 and 8. In the absence of observations, the HIRLAM analysis followed the forecast. Missing data in the time series close to freeze-up and break-up are due to missing observations, hence missing information in the feedback files (see Section 2.2). Differences between the years due to the different prevailing weather conditions are seen in the temperature variations.

Generally, FLake tended to melt the lakes too early in spring, as already indicated by the LID statistics (Section 4.2). The too early break-up and too warm LSWT in summer show up clearly in Kilpisjärvi (Figure 8). In Lappajärvi (Figure 9), the model and analysis were able to follow even quite large and quick variations of LSWT in summer, but tended to somewhat overestimate the maximum temperatures. Overestimation of the maximum temperatures by FLake was still more prominent in shallow lakes (not shown). In autumn over Lakes Lappajärvi and Kilpisjärvi, the forecasts and analyses followed closely the LSWT observations and reproduced the freeze-up dates within a few days, which was also typical to the majority of lakes.

Figure 10 shows a comparison of forecast and observed evolution of ice thickness and snow depth on Lappajärvi, Kilpisjärvi and Simpelejärvi in winter 2012-2013, typical also for the other lakes and years studied. The most striking feature is that there was no snow in the HIRLAM forecast.

On all three lakes, the ice thickness started to grow after freeze-up both according to the forecast and the observations. In the beginning HIRLAM ice grew faster than observed. However, according to the forecast ice thickness started to decrease in March of every year but according to the observations only a month or two later.

The too early break-up of lake ice in the absence of snow can be explained by the wrong absorption of the solar energy in the model. In reality, the main factor of snow and ice melt in spring is the increase of daily solar radiation. In HIRLAM, the downwelling short-wave irradiance at the surface is known to be reasonable, with some overestimation of the largest clear-sky fluxes and all cloudy fluxes (Rontu et al., 2017). Over lakes, HIRLAM uses constant values for the snow and ice shortwave reflection, with albedo values of 0.75 and 0.5, correspondingly. When there was no snow, the lake surface was thus assumed too dark. 25 % more absorption of an assumed maximum solar irradiance of 500 $\mathrm{Wm}^{-2}$ (valid for the latitude of Lappajärvi in the end of March) would mean availability of extra 125 $\mathrm{Wm}^{-2}$ for melting of the ice, which corresponds the magnitude of increase of available maximum solar energy within a month at the same latitude.

The forecast of too thick ice can also be explained by the absence of snow in the model. When there is no insulation by the snow layer, the longwave cooling of the ice surface in clear-sky conditions is more intensive and leads to faster growth of ice compared to the situation of snow-covered ice. In nature, ice growth can also be due to the snow transformation, a process whose parametrization in the models is demanding (Yang et al., 2013; Cheng et al., 2014).

Also the downwelling longwave radiation plays a role in the surface energy balance. We may expect values from 150 $\mathrm{Wm}^{-2}$ to 400 $\mathrm{Wm}^{-2}$ in the Nordic spring conditions, with the largest values related to cloudy and the smallest to clear-sky situations. The standard deviation of the predicted by HIRLAM downwelling longwave radiation fluxes has been shown to be of the order of 20 $\mathrm{Wm}^{-2}$, with a positive systematic error of a few $\mathrm{Wm}^{-2}$ (Rontu et al., 2017). Compared to the systematic effects related to absorption of the solar radiation, the impact of the longwave radiation variations on lake ice evolution is presumably small.

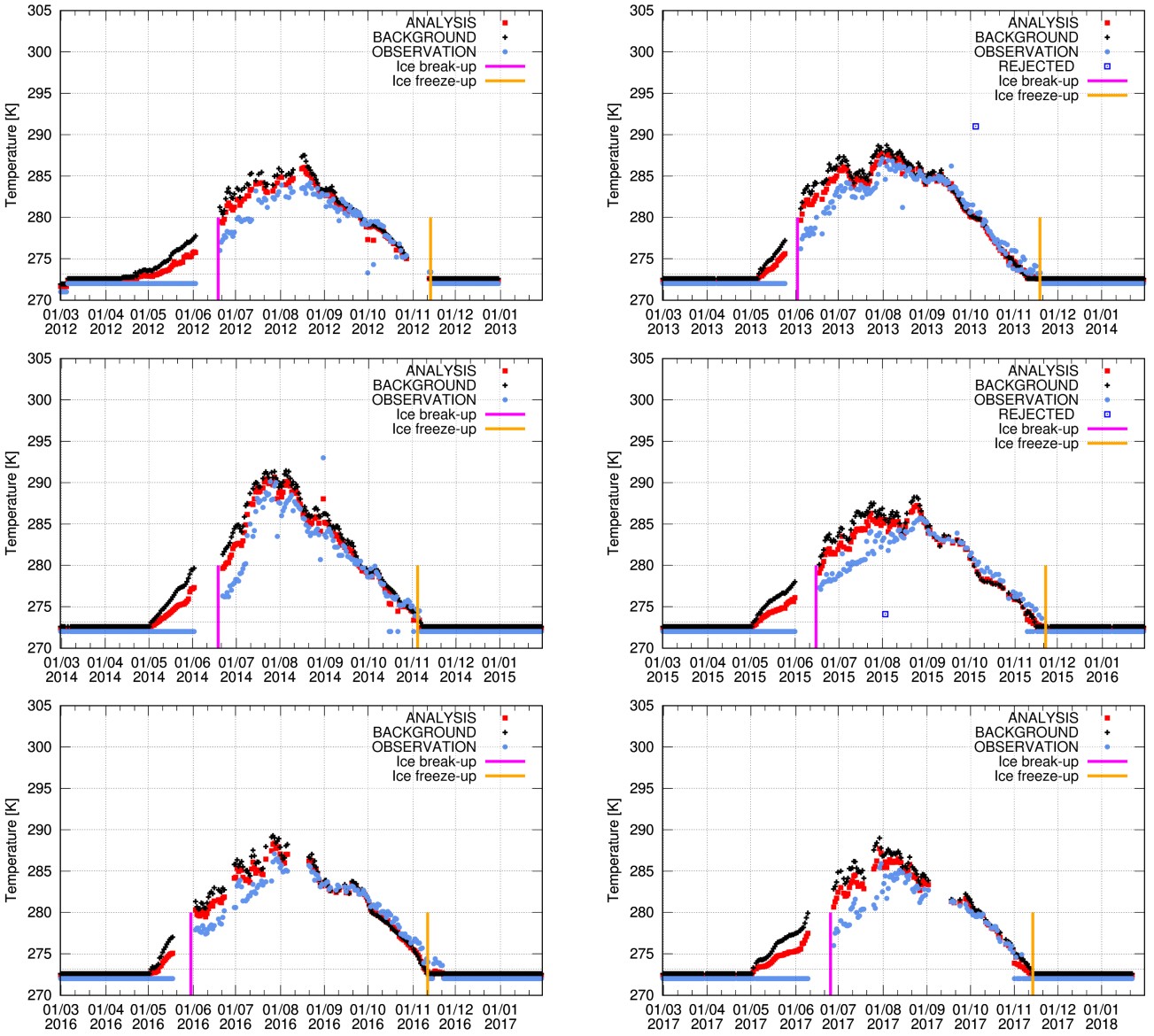

**Figure 8.** Time-series of the observed, analysed and forecast LSWT at the Kilpisjärvi observation location 20.82 E, 69.01 N. for the years 2012-2018 based on 06 UTC data. Markers are shown in the inserted legend. Observed freeze-up date (blue) and break-up date (red) are marked with vertical lines.

## 5   Discussion: snow on lake ice

The most striking result reported in Section 4 was the too early melting of the lake ice predicted by FLake in HIRLAM as compared to observations. We suggested that the early break-up is related to the missing snow on lake ice in HIRLAM. It was detected that a too large critical value to diagnose snow existence prevented practically all accumulation of the forecast snowfall on lake ice in the reference HIRLAM v.7.4, used operationally at FMI.

In general, handling of the snow cover on lake and sea ice is a demanding task for the NWP models. In HIRLAM,

snow depth observations are included into the objective analysis over the land areas, but not over ice where no observations are widely available in real time. Over land, snow depth and temperature are treated prognostically using dedicated parametrizations (in HIRLAM, similar to Samuelsson et al., 2006, 2011, see also Boone et al., 2017). Over the sea, a simple prognostic parametrization of sea ice temperature is applied in HIRLAM but neither the thickness of ice nor the depth or temperature of snow on ice are included (Samuelsson et al., 2006). Batrak et al. (2018) provide a useful review and references concerning prognostic sea ice schemes and their snow treatment in NWP models. An essential difference

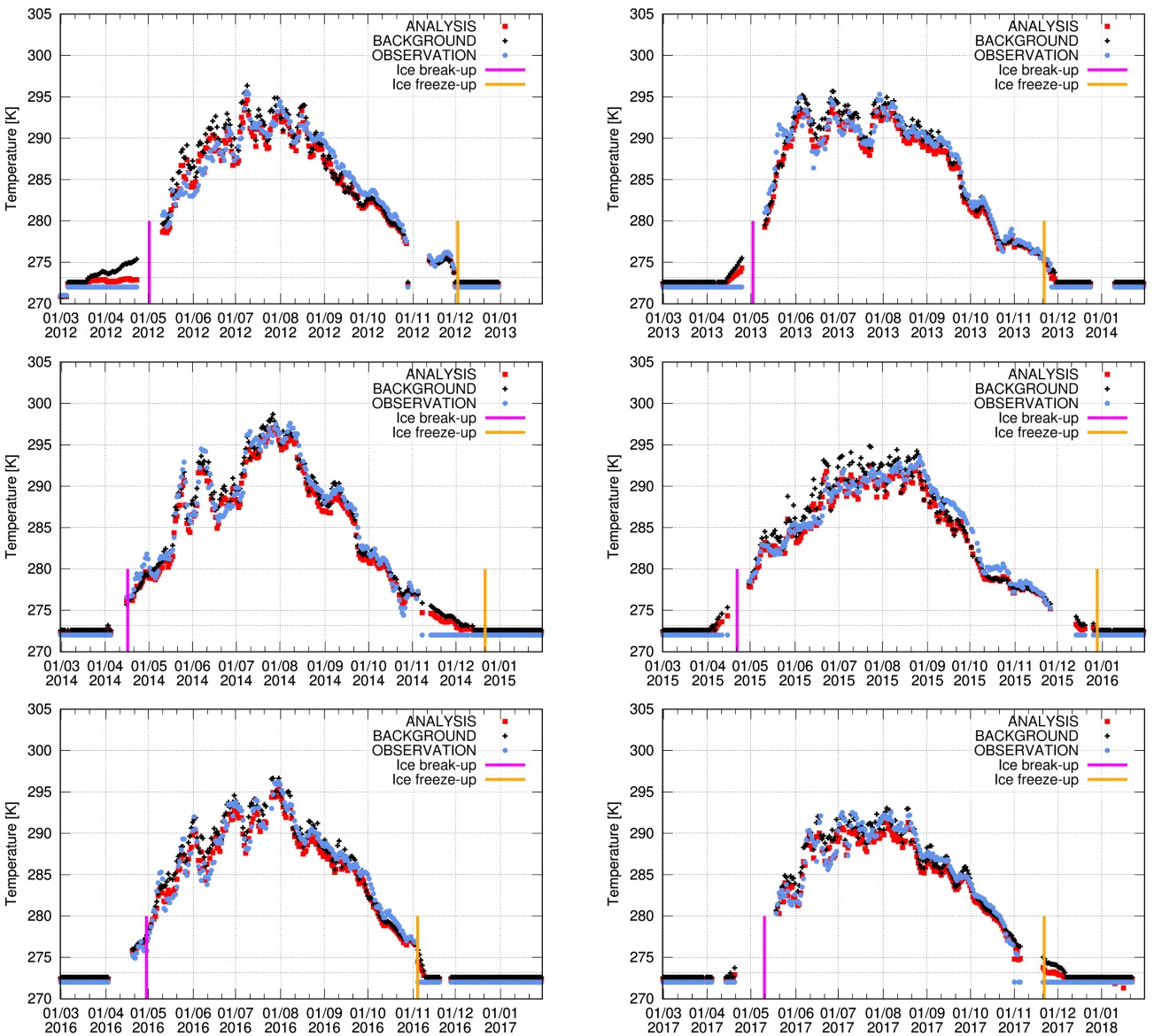

**Figure 9.** As for Figure 8 but for lake Lappajärvi, 23.67 E, 63.15 N

between the simple sea ice scheme and the lake ice scheme applied in HIRLAM is that the former relies on external data on the existence of sea ice cover, provided by the objective analysis, while the latter includes prognostic treatment of the lake water body also. This means that the lake ice freezes and melts in the model depending on the thermal conditions of lake water, evolving throughout the seasons.

The ice thickness, snow depth and ice and snow temperatures are prognostic variables of FLake. When the FLake parametrizations were introduced into HIRLAM (Kourzeneva et al., 2008; Eerola et al., 2010), parametrization of the snow thickness and snow temperature was first excluded. In the COSMO NWP model, snow is implicitly accounted for by modifying ice albedo using empirical data

on its temperature dependence (Mironov et al., 2010). This way was applied also e.g. in a recent study over the Great Lakes (Baijnath-Rodino and Duguay, 2019).

Semmler et al. (2012) performed a detailed winter-time comparison between FLake and a more complex snow and ice thermodynamic model (HIGHTSI) on a small lake in Alaska. FLake includes only one ice and one soil layer, while HIGHTSI represents a more advanced multilayer scheme. Atmospheric forcing for the stand-alone experiments was provided by HIRLAM. Based on their sensitivity studies, Semmler et al. (2012) suggested three simplifications to the original, time-dependent snow-on-ice parametrizations of FLake: use a prescribed constant snow density, modify the value of the prescribed molecular heat conductivity and use

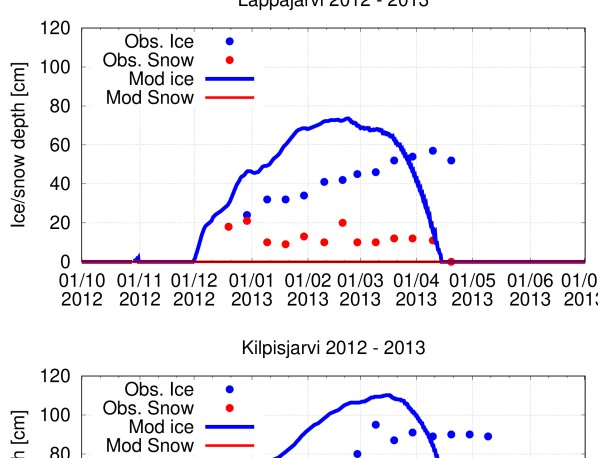

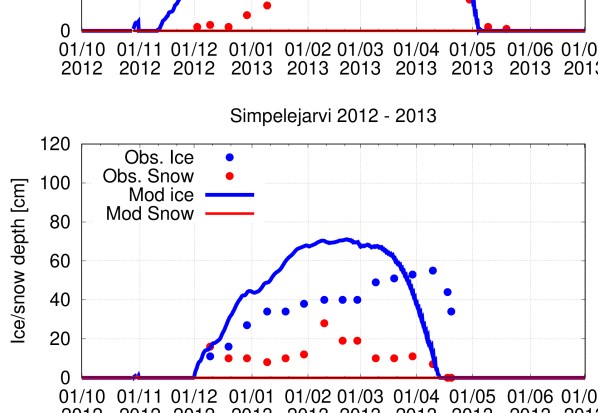

**Figure 10.** Evolution of ice (blue) and snow (red) thickness at Lakes Lappajärvi, Kilpisjärvi and Simpelejärvi during winter 2012-2013.

prescribed constant albedos of dry snow and ice. Later, a similar comparison was performed over Lake Kilpisjärvi (Yang et al., 2013), confirming the improvements due to the updated snow parametrizations in FLake. Implementation of these modifications allowed to include the parametrization of snow on lake ice also into HIRLAM (Section 2.1).

In FLake, snow on lake ice is accumulated from the predicted snowfall. Snow melt on lake ice is related to snow and ice temperatures. In case of FLake integrated into HIRLAM, accumulation and melt are updated at every time step of the advancing forecast. Very small amounts of snow are considered to fall beyond the accuracy of parametrizations and are removed. This is controlled by a critical limit, which was set too large (one millimeter instead of ten micrometers) in HIRLAM v.7.4. Due to the incorrect critical value, practically no snow accumulated on lake ice in the FMI operational HIRLAM, validated in this study. In a HIRLAM test experi-

ment, where the original smaller value was used, up to 17 cm of snow accumulated on lake ice within a month (Janurary 2012, not shown).

## 6    Conclusions and outlook

In this study, *in-situ* lake observations from the Finnish Environment Institute were used for validation of the HIRLAM NWP model, which is applied operationally in the Finnish Meteorological Institute. HIRLAM contains Freshwater Lake prognostic parametrizations and an independent objective analysis of lake surface state. We focused on comparison of observed and forecast lake surface water temperature, ice thickness and snow depth in the years 2012 - 2018. Because the HIRLAM system was unmodified during this period, a long uniform dataset was available for evaluation of the performance of FLake integrated into an operational NWP model. On the other hand, no conclusions about the impact of the lake surface state on the operational forecast of the near-surface temperatures, cloudiness or precipitation can be drawn because of the lack of alternative forecasts (without FLake) for comparison.

On average, the forecast and analysed LSWT were warmer than observed with systematic errors of 0.91 K and 0.35 K, correspondingly. The mean absolute errors were 1.94 and 1.32 K. Thus, the independent observation-based analysis of *in-situ* LSWT observations was able to improve the FLake +6 h forecast used as the first guess. However, the resulting analysis is by definition not used for correction of the FLake forecast but remains an independent by-product of HIRLAM. It appeared that FLake LSWT as well as the analysed LSWT can follow quite large and quick variations of LSWT in summer. However, an overestimation of the FLake LSWT summer maxima was found, especially for the shallow lakes. This behaviour of FLake is well known, documented earlier e.g. by Kourzeneva (2014). It arises due to the difficulty to handle correctly the mixing in the near-surface water layer that is intensively heated by the sun. In HIRLAM-FLake, the direct radiative heating of the bottom sediments is not taken into account, which may also contribute to this error.

Forecast freeze-up dates were found to correspond the observations well, typically within a week. The forecast ice thickness tended to be overestimated, still the break-up dates over most of the lakes occured systematically several weeks too early. Practically no forecast snow was found on the lake ice, although the snow parametrization by FLake was included in HIRLAM. The reason for the incorrect behaviour was related to a too large critical value to diagnose snow existence that prevented the accumulation of snow on lake ice. The too early melting and overestimated ice thickness differ from the results by Pietikäinen et al. (2018); Yang et al. (2013); Kourzeneva (2014), who reported somewhat too late melting of the Finnish lakes when FLake with realistic snow parametrizations was applied within a climate

model or stand-alone driven by NWP data. It can be concluded that a realistic parametrization of snow on lake ice is important in order to describe correctly the lake surface state in spring.

Small lakes and those of complicated geometry cause problems for the relatively coarse HIRLAM grid of seven-kilometre horizontal resolution. The problems are related to the observation usage, forecast and validation, especially when interpolation and selection of point values are applied. The observations and model represent different spatial scales. For example, the comparison of the freeze-up and break-up dates was based on diagnostics of single-gridpoint values that were compared to observations which represent entire lakes as overseen from the observation sites. Also the results of LID diagnostics were sensitive to the criteria for definition of the ice existence in HIRLAM. All this adds unavoidable inaccuracy into the model-observation intercomparison but does not change the main conclusions of the present study.

SYKE LSWT observations used for the real-time analysis are regular and reliable but do not always cover the days immediately after break-up or close to freeze-up. This is partly because the quality control of HIRLAM LSWT analysis utilizes the SYKE statistical lake water temperature model results in a strict way. Although the 27 observations are located all over the country, they cover a very small part of the lakes and their availability is limited to Finland. SYKE observations of the ice and snow depth as well as the freeze-up and break-up dates provide valuable data for the validation purposes but not for the real-time analysis.

A need for minor technical corrections in the FMI HIRLAM system was revealed. The coefficient influencing snow accumulation on lake ice was corrected based on our findings. Further developments and modifications are not foreseen because the HIRLAM NWP systems, applied in some European weather services, are being replaced by kilometre-scale ALADIN-HIRLAM forecasting systems (Termonia et al., 2018; Bengtsson et al., 2017), where the prognostic FLake parametrizations are also available. This system uses the newest version of the global lake database (GLDB v.3) and contains updated snow and ice properties. The objective analysis of lake surface state is yet to be implemented, taking into account the HIRLAM experience summarized in this study and earlier by Kheyrollah Pour et al. (2017). In the future, an important source of wider observational information on lake surface state are the satellite measurements, whose operational application in NWP models still requires further work.

*Code and data availability.* Observational data was obtained from SYKE open data archive SYKE, 2018 as follows: LID was fetched 15.8.2018, snow depth 17.9.2018 and ice thickness 16.10.2018 from http://rajapinnat.ymparisto.fi/api/Hydrologiarajapinta/1.0/odataquerybuilder/. A supplementary file containing the freeze-up and break-up dates as picked and prepared for the lakes studied here is attached. Data picked from HIRLAM archive are attached as supplementary files: data from the objective analysis feedback files (observed, analysed, forecast LSWT interpolated to the 27 active station locations) and from the gridded output of the HIRLAM analysis (analysed LSWT, forecast ice and snow thickness from the nearest gridpoint of all locations used in the present study).

In this study, FMI operational weather forecasts resulting from use of HIRLAM v.7.4 (rc1, with minor local updates) were validated against lake observations. The HIRLAM reference code is not open software but the property of the international HIRLAM-C programme. For research purposes, the code can be requested from the programme (hirlam.org). The source code of the version operational at FMI, relevant for the present study, are available from the authors upon request.

*Author contributions.* Laura Rontu computed the LSWT statistics based on HIRLAM feedback files. Kalle Eerola performed the freeze-up and break-up date, snow and ice thickness comparisons based on data picked from HIRLAM grib files. Matti Horttanainen prepared observation data obtained via SYKE open data interface and lake depths from GLDB v.3. Laura Rontu composed the manuscript text based on input from all authors.

*Competing interests.* No competing interests are present.

*Acknowledgements.* Our thanks are due to Joni-Pekka Pietikäinen and Ekaterina Kourzeneva for discussions and information, to Margarita Choulga and Olga Toptunova for the support with the updated GLDB v.3 data, and to Emily Gleeson for advice with English language. The comments of three anonymous reviewers and the editor helped significantly to improve the presentation of our results.

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

**Table A1.** Prognostic and diagnostic lake variables within HIRLAM

| variable | unit | type |
|---|---|---|
| temperature of snow on lake ice | K | prog by FLake |
| temperature of lake ice | K | prog by FLake |
| mean water temperature | K | prog by FLake |
| mixed layer temperature | K | prog by FLake |
| bottom temperature | K | prog by FLake |
| mixed layer depth | m | prog by FLake |
| thermocline shape factor | - | prog by FLake |
| lake ice thickness | m | prog by FLake |
| snow depth on lake ice | m | prog by FLake |
| (temperature of upper layer sediments | K | prog by FLake) |
| (thickness of upper layer sediments | m | prog by FLake) |
| LSWT | K | diag by FLake |
| | | = mixed layer temperature if no ice |
| lake surface temperature | K | diag by FLake |
| | | uppermost temperature: LSWT or ice or snow |
| LSWT | K | anal by HIRLAM |
| | | flag value 272 K when there is ice |
| fraction of lake ice | [0-1] | diag fraction in HIRLAM grid |
| lake surface roughness | m | diag by HIRLAM |
| screen level temperature over lake | K | diag by HIRLAM |
| screen level abs.humidity over lake | $kgkg^{-1}$ | diag by HIRLAM |
| anemometer level u-component over lake | $ms^{-1}$ | diag by HIRLAM |
| anemometer level v-component over lake | $ms^{-1}$ | diag by HIRLAM |
| latent heat flux over lake | $Wm^{-2}$ | diag by HIRLAM |
| sensible heat flux over lake | $Wm^{-2}$ | diag by HIRLAM |
| scalar momentum flux over lake | Pa | diag by HIRLAM |
| SW net radiation over lake | $Wm^{-2}$ | diag by HIRLAM |
| LW net radiation over lake | $Wm^{-2}$ | diag by HIRLAM |
| depth of lake | m | prescr in HIRLAM grid |
| fraction of lake | [0-1] | prescr in HIRLAM grid |

Denotation: prog = prognostic, diag = diagnostic, prescr = prescribed, anal = result of OI. Bottom sediment calculations by FLake are not applied in HIRLAM.

**Table A2.** Lakes with SYKE observations used in this study.

| NAME | LON | LAT | MEAND (m) | MAXD (m) | AREA (kgm$^{-2}$) | HIRD (m) | HIRFR | HIRID |
|---|---|---|---|---|---|---|---|---|
| Pielinen | 29.607 | 63.271 | 10.1 | 61.0 | 894.2 | 10.0 | 0.916 | 4001 |
| Kallavesi | 27.783 | 62.762 | 9.7 | 75.0 | 316.1 | 10.0 | 0.814 | 4002 |
| Haukivesi | 28.389 | 62.108 | 9.1 | 55.0 | 560.4 | 10.0 | 0.725 | 4003 |
| Saimaa | 28.116 | 61.338 | 10.8 | 85.8 | 1,377.0 | 10.0 | 0.950 | 4004 |
| Pääjärvi1 | 24.789 | 62.864 | 3.8 | 14.9 | 29.5 | 3.0 | 0.430 | 4005 |
| Nilakka | 26.527 | 63.115 | 4.9 | 21.7 | 169.0 | 10.0 | 0.866 | 4006 |
| Konnevesi | 26.605 | 62.633 | 10.6 | 57.1 | 189.2 | 10.0 | 0.937 | 4007 |
| Jääsjärvi | 26.135 | 61.631 | 4.6 | 28.2 | 81.1 | 10.0 | 0.750 | 4008 |
| Päijänne | 25.482 | 61.614 | 14.1 | 86.0 | 864.9 | 10.0 | 0.983 | 4009 |
| Ala-Rieveli | 26.172 | 61.303 | 11.3 | 46.9 | 13.0 | 10.0 | 0.549 | 4010 |
| Kyyvesi | 27.080 | 61.999 | 4.4 | 35.3 | 130.0 | 10.0 | 0.810 | 4011 |
| Tuusulanjärvi | 25.054 | 60.441 | 3.2 | 9.8 | 5.9 | 3.0 | 0.174 | 4012 |
| Pyhäjärvi | 22.291 | 61.001 | 5.5 | 26.2 | 155.2 | 5.0 | 0.922 | 4013 |
| Längelmävesi | 24.370 | 61.535 | 6.8 | 59.3 | 133.0 | 10.0 | 0.875 | 4014 |
| Pääjärvi2 | 25.132 | 61.064 | 14.8 | 85.0 | 13.4 | 14.0 | 0.350 | 4015 |
| Vaskivesi | 23.764 | 62.142 | 7.0 | 62.0 | 46.1 | 10.0 | 0.349 | 4016 |
| Kuivajärvi | 23.860 | 60.786 | 2.2 | 9.9 | 8.2 | 10.0 | 0.419 | 4017 |
| Näsijärvi | 23.750 | 61.632 | 14.7 | 65.6 | 210.6 | 10.0 | 0.850 | 4018 |
| Lappajärvi | 23.671 | 63.148 | 6.9 | 36.0 | 145.5 | 10.0 | 1.000 | 4019 |
| Pesiöjärvi | 28.650 | 64.945 | 3.9 | 15.8 | 12.7 | 7.0 | 0.290 | 4020 |
| Rehja-Nuasjärvi | 28.016 | 64.184 | 8.5 | 42.0 | 96.4 | 10.0 | 0.534 | 4021 |
| Oulujärvi | 26.965 | 64.451 | 6.9 | 35.0 | 887.1 | 10.0 | 1.000 | 4022 |
| Ounasjärvi | 23.602 | 68.377 | 6.6 | 31.0 | 6.9 | 10.0 | 0.166 | 4023 |
| Unari | 25.711 | 67.172 | 5.0 | 24.8 | 29.1 | 10.0 | 0.491 | 4024 |
| Kilpisjärvi | 20.816 | 69.007 | 19.5 | 57.0 | 37.3 | 22.0 | 0.399 | 4025 |
| Kevojärvi | 27.011 | 69.754 | 11.1 | 35.0 | 1.0 | 10.0 | 0.016 | 4026 |
| Inarijärvi | 27.924 | 69.082 | 14.3 | 92.0 | 1,039.4 | 14.0 | 0.979 | 4027 |
| Simpelejärvi | 29.482 | 61.601 | 9.3 | 34.4 | 88.2 | 10.0 | 0.548 | 40241 |
| Pökkäänlahti | 27.264 | 61.501 | 8.0 | 84.3 | 58.0 | 10.0 | 0.299 | 40261 |
| Muurasjärvi | 25.353 | 63.478 | 9.0 | 35.7 | 21.1 | 10.0 | 0.060 | 40263 |
| Kalmarinselkä | 25.001 | 62.786 | 5.7 | 21.9 | 7.1 | 5.0 | 0.330 | 40271 |
| Summasjärvi | 25.344 | 62.677 | 6.7 | 40.5 | 21.9 | 10.0 | 0.555 | 40272 |
| Iisvesi | 27.021 | 62.679 | 17.2 | 34.5 | 164.9 | 18.0 | 0.456 | 40277 |
| Hankavesi | 26.826 | 62.614 | 7.0 | 49.0 | 18.2 | 18.0 | 0.100 | 40278 |
| Petajävesi | 25.173 | 62.255 | 4.2 | 26.6 | 8.8 | 3.0 | 0.245 | 40282 |
| Kukkia | 24.618 | 61.329 | 5.2 | 35.6 | 43.9 | 10.0 | 0.299 | 40308 |
| Ähtärinjärvi | 24.045 | 62.755 | 5.2 | 27.0 | 39.9 | 10.0 | 0.266 | 40313 |
| Kuortaneenjärvi | 23.407 | 62.863 | 3.3 | 16.2 | 14.9 | 10.0 | 0.277 | 40328 |
| Lestijärvi | 24.716 | 63.584 | 3.6 | 6.9 | 64.7 | 10.0 | 0.513 | 40330 |
| Pyhäjärvi | 25.995 | 63.682 | 6.3 | 27.0 | 121.8 | 10.0 | 0.266 | 40331 |
| Lentua | 29.690 | 64.204 | 7.4 | 52.0 | 77.8 | 7.0 | 0.600 | 40335 |
| Lammasjärvi | 29.551 | 64.131 | 4.3 | 21.0 | 46.8 | 3.0 | 0.200 | 40336 |
| Naamankajärvi | 28.246 | 65.104 | 2.9 | 14.0 | 8.5 | 7.0 | 0.299 | 40342 |
| Korvuanjärvi | 28.663 | 65.348 | 6.0 | 37.0 | 15.4 | 10.0 | 0.342 | 40343 |
| Oijärvi | 25.930 | 65.621 | 1.1 | 2.4 | 21.0 | 10.0 | 0.333 | 40345 |

Denotation: LON and LAT are the longitude E and latitude N in degrees, MEAND and MAXD are the mean and maximum depths and AREA is the water surface area from the updated lake list of GLDB v.3 (Margarita Choulga, personal communication), HIRD and HIRFR are the mean lake depth and fraction of lakes [0...1] interpolated to the selected HIRLAM gridpoint, taken from the operational HIRLAM that uses GLDB v.2 as the source for lake depths. HIRID is the lake index used by HIRLAM and in this study. Above the middle line are the 27 lakes with both LSWT and LID observations, below the 18 lakes where only LID was available.