# Peer review of "Validation of lake surface state in the HIRLAM v.7.4 NWP model against *in-situ* measurements in Finland"

_Geoscientific Model Development, 2018_

## Short Comment (SC1) · 7 Nov 2018

Dear authors,

In my role as Executive editor of GMD, I would like to bring to your attention our Editorial version 1.1:

http://www.geosci-model-dev.net/8/3487/2015/gmd-8-3487-2015.html

This highlights some requirements of papers published in GMD, which is also available on the GMD website in the 'Manuscript Types' section:

http://www.geoscientific-model-development.net/submission/manuscript_types.html

In particular, please note that for your paper, the following requirements have not been

met in the Discussions paper:

- "The main paper must give the model name and version number (or other unique identifier) in the title."

- "All papers must include a section, at the end of the paper, entitled 'Code availability'. Here, either instructions for obtaining the code, or the reasons why the code is not available should be clearly stated. It is preferred for the code to be uploaded as a supplement or to be made available at a data repository with an associated DOI (digital object identifier) for the exact model version described in the paper. Alternatively, for established models, there may be an existing means of accessing the code through a particular system. In this case, there must exist a means of permanently accessing the precise model version described in the paper. In some cases, authors may prefer to put models on their own website, or to act as a point of contact for obtaining the code. Given the impermanence of websites and email addresses, this is not encouraged, and authors should consider improving the availability with a more permanent arrangement. After the paper is accepted the model archive should be updated to include a link to the GMD paper."

Therefore please add the version number of HIRLAM to the title of your manuscript upon revision. This is important also for evaluation paper, as the evaluation result can differ for different model versions. Additionally, add the information how this HIRLAM version can be accessed.

Yours,

Astrid Kerkweg

---

## Referee Comment (RC1) · Anonymous Referee #1 · 12 Dec 2018

General comments:

The paper presents results of HIRLAM (v7.4) model integrated to Flake model, lake surface state validation against in-situ observations of lake water temperature and ice cover during the period of 2012-2018 in Finland. In general, the paper structure is good and it is mainly written well. Same validation results against these in-situ observations have not been published earlier, eventhough some earlier papers have dealt with lake temperature and ice cover observation use in the HIRLAM. However, the noticed bug related to ice cover modelling is rather fundamental in physical way, and dominating the results, and makes me consider revising results with proper snowfall calculations on ice. It seems that in the future the HIRLAM model is no longer used and will be substituted with a new model. In that aspect, erroneous calculation could be docu-

mented in this article. The figures and tables could be improved and should be made more visual and reader-friendly; I will provide some specific comments on them. Especially figures and tables should run better in line with the text. Now, some figures are mentioned many pages before that they appear.

Specific comments:

1. Introduction, first paragraph (page 1, lines 16-19): Please provide some references.

2. You have used observations data for the year 2018 eventhough it is current year, usually provisional data. Is the in-situ data used in the analysis quality controlled? When the in-situ data was uploaded? And until which date the year 2018 data are used?

3. Page 3, Figure 1. I would like to have it more visual-friendly. Is there certain meaning with arrow line thickness, if not then harmonize.

4. Page 5, line 16. Please make reference to SYKE network, which year status it is? (There are 34 sites in the network in year 2018 according to the SYKE database)

5. Chapter 3.2.2. Freezing and melting dates. Article Korhonen (2006) has introduced terms for freezing and break-up in English, please use those. See: Korhonen, J. 2006. Long-term changes in lake ice cover in Finland. Nordic Hydrology 37(4-5): 347–363.

6. Please state little bit more why these lakes were chosen. Were they only ones large enough to HIRLAM grid or were there other criteria?

7. I suggest combining figures 3 & 4 to same gridded figure with four graphs. Remove from temperature scale dots after the kelvin numbers. In figure caption open up meaning of fc an fob, an.

8. Chapter 4.2. is little bit hard to read/understand. Try to rewrite it more clear.

9. Page 10: Text paragraph, it is not clear what are different percentage categories.

10. Table 2: What are the units in this table?

11. I suggest combining Figures 5 and 6 to same figure (a and b)

12. Page 12, last paragraph: make more clear in the text if you are talking about HIRLAM (analysed/forecast) or observed freezing and melting days.

13. Chapter 4.3. Make a reference to where lake area/depth records are taken. GLDB perhaps?

14. Figures 7-10 could be combined to one gridded figure (a, b, c, d) Remove dots after Kelvin scale numbers.

15. Figure 11. & 12. Add variable name and Unit in Y-scale. Just one legend could be below graphs since they are all same. For codes 28 and 29 use verbal definitions, please. It seems data until early 2018 is used?

16. Figure 13. Add variable name and Unit in the Y-scale. In headings, use only lake name and years: Lappajärvi 2012-2013, Kilpisjärvi 2012-2013, Simpelejärvi 2012-2013.

Technical/typo corrections:

1) Abstract: line 3 "integrated to HIRLAM" -> integrated into HIRLAM

2) Use wording "in-situ" or "in situ" through whole text. Now there are both versions in the text.

3) I would use "lake ice freezing and "lake ice melting" instead of lake freezing and melting (all text) (e.g. page 2, line 21)

4) Page 2, line 31: I would consider revising wording "became available"

5) Page 4, line 31: I would consider revising wording "basic material"

6) Figure 2. Page 6: Please note that abbreviation LID has not been introduced in the text before.

[Figure]

7) Chapter 3.2.2 "codes 27-30" should not be used in the text or figures, use instead verbal definitions. Codes are so called database codes and not normally used as definitions. They are irrelevant as code numbers.

8) Please check through the text that LWT and LSWT are used coherently. Page 13, line 1: LWST -> LSWT, Page 18, line 13 SYKE LSWT?

9) Chapter 3.2.3 Ice thickness and snow depth on lakes

10) Page 7, line 8: typo Simpelejärvi

11) Chapter 3.3.1. Lake surface water temperature (LSWT)

12) Page 8, line 2: Use verbal definition instead of category 29. Same in line 3 for category 28.

13) Page 8, line 9: SYKE LWT observations

14) Page 8, line 21: typo known

15) Page 15, line 9: 125 Wm-2 (superscript)

16) Page 15, line 19: 2012-2018

17) Page 18, line 1: wrong -> incorrect/erroneous

18) Page 18, line 17: ice thickness and snow depth

19) References: Please check that all references are formatted same way. For example, if many initial letters using space between or not in consistent way. I noticed some typos:

Page 24,1. Potes, M. -> Potes, M.

Page 24, line 22. Gandin, L. missing :

Page 25, line 5. Remove ++ after Hydrology Research.

[Figure]

Page 25, line 11. co authors -> write all names

Page 25, line 33 et al > write all names

Page 25, line 33. Yang, Y., coauthors -> write all names and put the year in the end

―――――――――――――――――――

---

## Referee Comment (RC2) · Anonymous Referee #2 · 12 Jan 2019

General comments:

Rontu et al. utilize archived forecast data (2012-2018) from the NWP model HIRLAM to validate the analysed and forecasted state of lakes with respect to observations within a model domain. Due to unfortunate circumstances this specific HIRLAM version included a bug which prevented snow to accumulate on the lake ice. Due to this bug the model data related to ice behaviour and spring LSWT temperature became unrealistic and therefore the corresponding results and discussions are of very limited interest. Okay, it illustrates the importance of representing snow on ice when simulating lakes in cold climate conditions.

The manuscript is in general carefully written and can be considered as a useful guidance on how to validate the state of lakes in a NWP or climate model when correspond-

ing in-situ observational data are available. The authors carefully describe uncertainties with respect to representativeness of observations and representation of lakes in a model domain. Also, they describe how ice conditions may be estimated based on other data. All this information can be valuable for scientists planning similar exercises for other combinations of model and lake observations.

As the authors say it is a well known behaviour of FLake to overestimate summer LSWT. This is also seen in the presented results. However, it can not be excluded that part of those biases presented may be explained by for example any biases in near-surface temperature conditions in general. After all, the lakes represent only some 10% of the land area even in Finland so a bias in near-surface air temperature due to discrepancies in representation of land processes can also contribute to the presented biases. Thus, I would like to see a comment on the general near-surface temperature bias in this HIRLAM setup. The authors do comment on the quality of simulated down-welling short-wave irradiance but a comment on long-wave would also be relevant.

Detailed comments:

Page 2, line 3: Sounds a bit strange to combine observed LSWT and simulated ice thickness to estimate fractional ice: " Fractional ice cover (lake ice concentration in each grid-square of the model) is estimated separately based on the analysed LSWT and the ice thickness predicted by FLake."

Page 5, line 15 19: Here you refer to Figure 2 for the first time but in the caption of Figure 2 you use the abbreviation LID which is defined later in the text. Please, e.g., introduce "lake ice dates" also in the figure caption for clarity.

Page 8, lines 1-2: A bit strangely formulated sentence: "including in the comparison data over all months". Please make it more clear.

Pages 9-12, Section 4.2: The bug which prevents snow to accumulate on ice in this HIRLAM version will seriously affect all results presented in this section so I think it

would be fair to the reader to comment on this in the beginning of this section although it has been mentioned in previous sections.

Page 13, line 5: You say that "Lake Kilpisjärvi is an Arctic lake at the elevation of 473 m". This is a complex terrain area so the height difference between the real lake and the model lake might contribute to estimated biases in temperature. What is the corresponding height of the HIRLAM grid box here? Would a height correction of temperature make any difference for the results?

Figure 1: In the text it says that (page 2, line 33 – page 3, line 1) "the prognostic Flake variables are not corrected using the analysed LSWT, which would require advanced data assimilation methods" but in the figure it says "INDEPENDENT LAKE DATA AS-SIMILATION IN AN INTEGRATED NWP + LAKE MODEL". I suggest to remove "DATA ASSIMILATION" here since that is not done according to the text. And ice cover is simply 0 or 1 when ice is present or not, right? So, this is not really a diagnostic estimation I would say. Or does this include something else which is not yet clear from the text. . .? Okay, becomes clear on page 4. Maybe better to refer to Figure 1 a bit later when the background to the figure is clear from the text.

Figure 11: Colour indications of freezing and melting dates in the caption (blue and red) do not fit with the figure (orange and magenta).

[Figure]

---

## Referee Comment (RC3) · Anonymous Referee #3 · 26 Jan 2019

General comments:

The paper presents the detailed validation of the FLake model implemented in the HIRLAM NWP system, focusing mainly on the lake surface state and utilizing in situ measurements. The validation period is considerably large spanning over six years and a large number of lakes are included in the investigation. The validation area covers only Finnish lakes, consequently results are referring to arctic conditions and might not be generalized to other climate regimes. The technical properties of the modelling system as well as the observational dataset are described properly. A lake water temperature assimilation scheme is also presented, however, it is mentioned that this is only a diagnostic product. Perhaps, the application areas of this product could be highlighted so that the purpose of it is clearer for the reader.

During the validation, lake surface water temperature (LSWT), freezing and melting dates and ice thickness are investigated. Regarding LSWT results are in line with previous studies, namely an overestimation by FLake is pointed out. Freezing dates are simulated by an adequate precision, however, melting dates are poorly forecasted. The cause of this problem is enlightened during the investigation of the ice and snow thicknesses, namely due to a coding error snow is not accumulated on the ice surface. Physical consequences of this bug (missing insulation in winter and different albedo in spring) are well described.

Detailed comments:

1. Page 5 line 18: it is mentioned that water temperature is measured at 20 cm below water surface. Could the authors comment, whether this depth was used also in previous validation studies they are referring to (e.g. Kourzeneva 2014). Also, are there any difficulties in the validation when water is already frozen, but ice thickness is not reaching 20 cm?

2. Page 10, line 8: "with an area of 1 km^-2" should be "with an area of 1 km^2"

3. Page 13 line 14: "common to the majority of lakes" is a bit vague, "similar to the results averaged over all lakes" might more precise.

4. Page 15, line 9: "125 Wm-2": "-2" should be superscript as one line above.

5. Perhaps the authors could shortly comment, whether the bug revealed had any detectable impact on the forecasts of atmospheric variables (e.g. 2 m temperature) in the HIRLAM model in the six year period.

6. The use of in-situ observations is definitely of great value in the validation of lake surface state, however, when describing plans the authors might comment on the usability of satellite products as well.

---

## Author Comment (AC1) · 20 Feb 2019

Thank you, we now mention the version number of HIRLAM in the title and explain code availability in the proper section.

---

## Author Comment (AC2) · 20 Feb 2019

Reply to reviewer 1

Thank you for your careful reading of the manuscript, leading to helpful remarks and suggestions, which we mostly agree with. We have made modifications throught the whole text, but the kept the line numbers of the original manuscript in this reply. Please find our detailed response below. The difference between our original and corrected manuscript versions is provided in an attached diffpdf file.

General comments: The paper presents results of HIRLAM (v7.4) model integrated to Flake model, lake surface state validation against in-situ observations of lake water temperature and ice cover during the period of 2012-2018 in Finland. In general, the

paper structure is good and it is mainly written well. Same validation results against these in-situ observations have not been published earlier, eventhough some earlier papers have dealt with lake temperature and ice cover observation use in the HIRLAM. However, the noticed bug related to ice cover modelling is rather fundamental in physical way, and dominating the results, and makes me consider revising results with proper snowfall calculations on ice. It seems that in the future the HIRLAM model is no longer used and will be substituted with a new model. In that aspect, erroneous calculation could be docu- mented in this article. The figures and tables could be improved and should be made more visual and reader-friendly; I will provide some specific comments on them. Espe- cially figures and tables should run better in line with the text. Now, some figures are mentioned many pages before that they appear.

Concerning the snow-on-ice bug, it has now been corrected in the operational HIRLAM system, that continues running at FMI. The coming spring will provide material to check if the melting of lake ice is better handled. Also, in earlier experiments described e.g. in Kheyrollah Pour et al., 2014 and Eerola et al., 2014, this bug was not present. However, the results in those experiments were not validated against the ice and snow thickness, even the ice dates were used to a limited extent. In these circumstances, we do not consider it useful to run new HIRLAM experiments for checking the impact of the correction. Please note that in the new operational NWP at FMI, based on HARMONIE-AROME, no lake observations are analysed but Flake runs as it would in a climate model, i.e. continuing directly from the previous short forecast.

We will come back to the figures and tables when replying the specific comments. We agree that they should be improved. To correct the setup of figures at distant pages (caused by use of the default latex with template in the manuscript mode) we will ask advice from the GMD typesetting specialists if needed.

Specific comments: 1. Introduction, first paragraph (page 1, lines 16-19): Please provide some references.

We have first of all added references to papers describing the influence of lakes on weather forecasting in general, then influence on NWP and finally importance of describing the existence of ice correctly. We have selected the references so that they contain further relevant references.

2. You have used observations data for the year 2018 eventhough it is current year, usually provisional data. Is the in-situ data used in the analysis quality controlled? When the in-situ data was uploaded? And until which date the year 2018 data are used?

The operational analysis uses LWT observations from SYKE in real time. Those are quality controlled by the HIRLAM optimal analysis system: 1) excluding each station and comparing interpolated to its location nearby observations and 2) comparison against first guess. We use these quality-checked values from analysis feedback files in this study. Possible corrections by SYKE, made afterwards, were not used. The LID data and ice and snow thickness observations were obtained from SYKE open data base for this study, relying on their quality control: LID was fetched 15.8.2018, snow depth 17.9.2018 and ice thickness 16.10.2018 from http://rajapinnat.ymparisto.fi/api/Hydrologiarajapinta/1.0/odataquerybuilder/

We added a sentence about the quality control and mention how the SYKE data was obtained.

3. Page 3, Figure 1. I would like to have it more visual-friendly. Is there certain meaning with arrow line thickness, if not then harmonize.

We now mention that the thin arrows are related to data flow between the HIRLAM analysis-forecast cycles while the thick ones describe processes within each cycle. We made also another correction to the Figure as suggested by reviewer 2.

4. Page 5, line 16. Please make reference to SYKE network, which year status it is? (There are 34 sites in the network in year 2018 according to the SYKE database) We

explain this better in section 3.2.1. , i.e. that there are 34 stations now from which we use in the operational HIRLAM the original year 2011 selection that has never been changed since that. Originally, we excluded rivers and a couple of stations that then seemed to send data less regularly. The list needs to be updated for HARMONIE-AROME if LSWT analysis will be introduced there in the future.

5. Chapter 3.2.2. Freezing and melting dates. Article Korhonen (2006) has introduced terms for freezing and break-up in English, please use those. See: Korhonen, J. 2006. Long-term changes in lake ice cover in Finland. Nordic Hydrology 37(4-5): 347–363. Thank you, we are aware of this terminology but selected freezing and melting according to the suggestion by our native linguistic advisor Emily Gleeson. In our earlier papers written together with our Canadian colleagues, we have used consistently the terms freeze-up and break-up. Now we did not like the suggested mixture of freezing and break-up, but perhaps there are good reasons to use this combination. We would like to leave the last word to our native British GMD editor of the current manuscript Jason Williams.

6. Please state little bit more why these lakes were chosen. Were they only ones large enough to HIRLAM grid or were there other criteria? The main criteria of selecting just these lakes for LID was the data availability: the most complete time series during the selected years, and a reasonable size that provided the best fraction of lake in HIRLAM grid. We now mention this.

7. I suggest combining figures 3 & 4 to same gridded figure with four graphs. Remove from temperature scale dots after the kelvin numbers. In figure caption open up meaning of fc an fob, an. We kept two figures, that we consider to be more clear in the final two-column setup of the journal. The fc-ob-an were added to captions.

8. Chapter 4.2. is little bit hard to read/understand. Try to rewrite it more clear. Thank you, we tried to clarify. This chapter is re-written totally to make it easier to read.

9. Page 10: Text paragraph, it is not clear what are different percentage categories.

Rewritten

10. Table 2: What are the units in this table? Thank you, units added

11. I suggest combining Figures 5 and 6 to same figure (a and b) Done

12. Page 12, last paragraph: make more clear in the text if you are talking about HIRLAM (analysed/forecast) or observed freezing and melting days. Rewritten

13. Chapter 4.3. Make a reference to where lake area/depth records are taken. GLDB perhaps? We renewed the list in Table A3 based on updated material for GLBD v.3 (not yet available at the Flake site but received by courtesy of Margarita Choulga), made the reference and mentioned it more clearly in the text.

14. Figures 7-10 could be combined to one gridded figure (a, b, c, d) Remove dots after Kelvin scale numbers. We created 2 figures and removed the dots.

15. Figure 11. & 12. Add variable name and Unit in Y-scale. Just one legend could be below graphs since they are all same. For codes 28 and 29 use verbal definitions, please. It seems data until early 2018 is used? Done. Data till summer is used 2018 (see above).

16. Figure 13. Add variable name and Unit in the Y-scale. In headings, use only lake name and years: Lappajärvi 2012-2013, Kilpisjärvi 2012-2013, Simpelejärvi 2012-2013. Done

Technical/typo corrections:

1) Abstract: line 3 "integrated to HIRLAM" -> integrated into HIRLAM Done

2) Use wording "in-situ" or "in situ" through whole text. Now there are both versions in the text. Done

3) I would use "lake ice freezing and "lake ice melting" instead of lake freezing and melting (all text) (e.g. page 2, line 21) Done

4) Page 2, line 31: I would consider revising wording "became available" "were obtained"

5) Page 4, line 31: I would consider revising wording "basic material" "is the basis of this study"

6) Figure 2. Page 6: Please note that abbreviation LID has not been introduced in the text before. This a setup problem, now we repeat the definition in the caption, too.

7) Chapter 3.2.2 "codes 27-30" should not be used in the text or figures, use instead verbal definitions. Codes are so called database codes and not normally used as definitions. They are irrelevant as code numbers. Replaced

8) Please check through the text that LWT and LSWT are used coherently. Page 13, line 1: LWST -> LSWT, Page 18, line 13 SYKE LSWT?

This is a bit problematic. Our idea was to call SYKE observations LWT because they are taken at the depth of 20 cm, not exactly at the surface that the satellite would have seen. However, Flake and HIRLAM analysis are dealing with LSWT even when the analysis is based on observed LWT. Perhaps the easiest solution is to call everything LSWT and mention the small difference when introducing the SYKE observations. We now did this.

9) Chapter 3.2.3 Ice thickness and snow depth on lakes Done

10) Page 7, line 8: typo Simpelejärvi Corrected

11) Chapter 3.3.1. Lake surface water temperature (LSWT) Corrected

12) Page 8, line 2: Use verbal definition instead of category 29. Same in line 3 for category 28. Done

13) Page 8, line 9: SYKE LWT observations See 8

14) Page 8, line 21: typo known Corrected

15) Page 15, line 9: 125 Wm-2 (superscript) Corrected

16) Page 15, line 19: 2012-2018 ? This is LaTex's work ...

17) Page 18, line 1: wrong -> incorrect/erroneous Corrected

18) Page 18, line 17: ice thickness and snow depth Corrected, also elsewhere

19) References: Please check that all references are formatted same way. For example, if many initial letters using space between or not in consistent way. I noticed some typos: Thank you, corrected as suggested

Page 24,1. Potes, M. -> Potes, M.

Page 24, line 22. Gandin, L. missing :

Page 25, line 5. Remove ++ after Hydrology Research.

Page 25, line 11. co authors -> write all names

Page 25, line 33 et al > write all names

Page 25, line 33. Yang, Y., coauthors -> write all names and put the year in the end

Reply to reviewer 2

Thank you for your helpful remarks and suggestions, which we mostly agree with. We have made modifications throught the whole text, but the kept the line numbers of the original manuscript in this reply. Please find our detailed response below. The difference between our original and corrected manuscript versions is provided in an attached diffpdf file.

General comments:

Rontu et al. utilize archived forecast data (2012-2018) from the NWP model HIRLAM to validate the analysed and forecasted state of lakes with respect to observations within a model domain. Due to unfortunate circumstances this specific HIRLAM version

included a bug which prevented snow to accumulate on the lake ice. Due to this bug the model data related to ice behaviour and spring LSWT temperature became unrealistic and therefore the corresponding results and discussions are of very limited interest. Okay, it illustrates the importance of representing snow on ice when simulating lakes in cold climate conditions.

Indeed, this bug was not present in our earlier experiments, e.g. Eerola et al., 2014 nor is it there in the latest (development) version of HIRLAM reference code. Now it is corrected also in the FMI operational version, that will allow to check the situation during the coming spring.

The manuscript is in general carefully written and can be considered as a useful guidance on how to validate the state of lakes in a NWP or climate model when corresponding in-situ observational data are available. The authors carefully describe uncertainties with respect to representativeness of observations and representation of lakes in a model domain. Also, they describe how ice conditions may be estimated based on other data. All this information can be valuable for scientists planning similar exercises for other combinations of model and lake observations.

Thank you for the positive comment, nice to hear that our methods are considered useful!

As the authors say it is a well known behaviour of FLake to overestimate summer LSWT. This is also seen in the presented results. However, it can not be excluded that part of those biases presented may be explained by for example any biases in near-surface temperature conditions in general. After all, the lakes represent only some 10% of the land area even in Finland so a bias in near-surface air temperature due to discrepancies in representation of land processes can also contribute to the presented biases. Thus, I would like to see a comment on the general near-surface temperature bias in this HIRLAM setup. The authors do comment on the quality of simulated downwelling short-wave irradiance but a comment on long-wave would also be relevant.

[Figure]

FLake works over the fraction of lake in each gridbox, driven by the average radiative and specific over-lake turbulent fluxes at the lake-atmosphere interface. The lake water and ice temperatures and other in-lake prognostic variables result from the Flake prognostic parametrizations. The resulting (diagnostic) LSWT represents the lake surface temperature in each gridbox, while the land-surface tile is taken care by other parametrizations (ISBA land-surface scheme), which also essentially solve the surface temperature from the equation of surface energy balance, taking into account also the heat conduction in ground. The grid-average screen-level temperature, that is regularly verified against observations, results from intelligent interpolation between the surface (e.g. LSWT) and the lowest model level temperature. In practice, the latter seems to dominate, but in principle, T2m could be wrong due to wrong LSWT but not vice versa. While there is no direct connection between the average (dominated by land surface) predicted surface temperature and LSWT, both might be inaccurate due to inaccurate atmospheric forcing. Wrong radiation transfer in the model, for example due to the cloudiness or incorrect handling of cloud-radiation interactions, biased near-surface air temperatures (at the lowest model level) or wrong turbulent fluxes in the atmospheric boundary layer could be sources of such inaccuracies.

Presumably, the shortwave radiation is the dominating factor for the lake water and ice thermodynamics during the year. In the equation of surface energy balance, the radiation fluxes are net fluxes over specific surface types, and these depend also on the prescribed surface properties, in our case e.g. on the lake ice and snow albedo. It would be worth while to perform sensitivity studies, e.g. with a single-column version of a NWP model, to reveal how Flake parametrizations would react to the inaccuracies of the atmospheric forcing and to quantify the related uncertaincies. This could be left for a further study for example in the framework of HARMONIE-AROME NWP system.

We added a sentence "Most importantly, FLake provides HIRLAM with the evolving lake surface (water, ice, snow) temperature, that influences the HIRLAM forecast of the grid-average near-surface temperatures." into the Flake description (Section 2.1). We

also discuss the uncertainties related to atmospheric forcing where only the shortwave flux is now mentioned in the conclusions. We come back to the temperature aspect in the reply of your Kilpisjärvi comment.

Detailed comments:

Page 2, line 3: Sounds a bit strange to combine observed LSWT and simulated ice thickness to estimate fractional ice: " Fractional ice cover (lake ice concentration in each grid-square of the model) is estimated separately based on the analysed LSWT and the ice thickness predicted by Flake." We improved our unfortunate formulation that allowed misunderstanding and relocated the explanations into their proper sections. There are two cases and two ways to estimate ice cover extent: in analysis, only LSWT exists, so it is used there in similar way that is done for SST – full ice concentration if the grid-average temperature is -0.5oC, none when it is 0oC and linearly in between. In the forecast by Flake, only ice thickeness is available. When it is larger than a small treshold value, the diagnostics decides that lakes existing in this gridbox are all frozen, i.e. the ice concentration is 1. There is a fraction of lakes in each gridbox, so the grid-scale ice fraction is obtained by multiplying the ice concentration with lake fraction. Thus, 'separately' meant: based on LSWT for analysis and based on ice thickness for forecast.

Page 5, line 15 19: Here you refer to Figure 2 for the first time but in the caption of Figure 2 you use the abbreviation LID which is defined later in the text. Please, e.g., introduce "lake ice dates" also in the figure caption for clarity. Done

Page 8, lines 1-2: A bit strangely formulated sentence: "including in the comparison data over all months". Please make it more clear. Done. The idea was that in the LSWT (obsa file) comparisons the winter months were excluded but here we used all data.

Pages 9-12, Section 4.2: The bug which prevents snow to accumulate on ice in this HIRLAM version will seriously affect all results presented in this section so I think it

would be fair to the reader to comment on this in the beginning of this section although it has been mentioned in previous sections. We now discuss the reasons for too early melting when showing the results here. This section has been largely rewritten.

Page 13, line 5: You say that "Lake Kilpisjärvi is an Arctic lake at the elevation of 473 m". This is a complex terrain area so the height difference between the real lake and the model lake might contribute to estimated biases in temperature. What is the corresponding height of the HIRLAM grid box here? Would a height correction of tem- perature make any difference for the results? The mean surface elevation of this gridbox where Lake Kilpisjärvi occupies around 40% of the area, is 614 m that is higher than the lake elevation because the lake is located in a valley surrounded by mountains. The diagnostic screen-level temperature, to which a heigh correction of temperature could be applied, plays no role in the model's air-surface energy exchange. To our under-standing, there is no way in Flake to apply height corrections as part of the prognostic calculations or diagnosis of lake surface (snow, ice, water) temperature, also we are not aware of studies related to this issue.

The observed LSWT is evidently measured on the lake at the correct height. During the objective analysis, Kilpisjärvi LSWT is influenced by the observation on the lake and possibly on the nearby lakes, which are probably too far from here to really influence the analysis result. Differences in lake elevations could in principle be taken into account in the optimal interpolation, but this is not currently done. More detailed discussion about the objective analysis of LSWT can be found in the paper by Kheyrollah Pour et al. 2017.

We now mention the difference in Kilpisjärvi and grid-average elevations.

Figure 1: In the text it says that (page 2, line 33 – page 3, line 1) "the prognostic Flake variables are not corrected using the analysed LSWT, which would require advanced data assimilation methods" but in the figure it says "INDEPENDENT LAKE DATA AS-SIMILATION IN AN INTEGRATED NWP + LAKE MODEL". I suggest to remove "DATA

ASSIMILATION" here since that is not done according to the text. And ice cover is simply 0 or 1 when ice is present or not, right? So, this is not really a diagnostic estimation I would say. Or does this include something else which is not yet clear from the text. . .? Okay, becomes clear on page 4. Maybe better to refer to Figure 1 a bit later when the background to the figure is clear from the text. We agree with the suggestion about "INDEPENDENT LAKE DATA ASSIMILATION" and replaced it with "OBJECTIVE ANALYSIS OF LSWT" in the figure. We also moved the figure and reference to it into Section 2.2.

Figure 11: Colour indications of freezing and melting dates in the caption (blue and red) do not fit with the figure (orange and magenta). Corrected

Reply to reviewer 3

Thank you for your helpful remarks and suggestions, which we mostly agree with. We have made modifications throught the whole text, but the kept the line numbers of the original manuscript in this reply. Please find our detailed response below. The difference between our original and corrected manuscript versions is provided in an attached diffpdf file.

General comments:

The paper presents the detailed validation of the FLake model implemented in the HIRLAM NWP system, focusing mainly on the lake surface state and utilizing in situ measurements. The validation period is considerably large spanning over six years and a large number of lakes are included in the investigation. The validation area covers only Finnish lakes, consequently results are referring to arctic conditions and might not be generalized to other climate regimes. The technical properties of the modelling system as well as the observational dataset are described properly. A lake water temperature assimilation scheme is also presented, however, it is mentioned that this is only a diagnostic product. Perhaps, the application areas of this product could be highlighted so that the purpose of it is clearer for the reader. We added a sentence

about the possible use of the diagnostic analysis into section 2.2.

During the validation, lake surface water temperature (LSWT), freezing and melting dates and ice thickness are investigated. Regarding LSWT results are in line with previous studies, namely an overestimation by FLake is pointed out. Freezing dates are simulated by an adequate precision, however, melting dates are poorly forecasted. The cause of this problem is enlightened during the investigation of the ice and snow thicknesses, namely due to a coding error snow is not accumulated on the ice surface. Physical consequences of this bug (missing insulation in winter and different albedo in spring) are well described.

Detailed comments:

1. Page 5 line 18: it is mentioned that water temperature is measured at 20 cm below water surface. Could the authors comment, whether this depth was used also in previous validation studies they are referring to (e.g. Kourzeneva 2014). Also, are there any difficulties in the validation when water is already frozen, but ice thickness is not reaching 20 cm? Yes, we have always used the same SYKE observations in our papers. These observations are only available during the ice-free period as mentioned in Section 3.2.1. and were used only then. There may be gaps between the observed freezing and melting dates and the dates when LSWT observations are made. Also, the locations of LID observations and LSWT measurements are not necessarily the same at the lakes where both types of observation are available. We added a couple of sentences about this into Section 3.2.2

2. Page 10, line 8: "with an area of 1 kmËĘ-2" should be "with an area of 1 kmËĘ2" Corrected

3. Page 13 line 14: "common to the majority of lakes" is a bit vague, "similar to the results averaged over all lakes" might more precise. Corrected as suggested

4. Page 15, line 9: "125 Wm-2": "-2" should be superscript as one line above. Corrected

5. Perhaps the authors could shortly comment, whether the bug revealed had any detectable impact on the forecasts of atmospheric variables (e.g. 2 m temperature) in the HIRLAM model in the six year period. The problem is that we do not know, because there is no way to compare the results with Flake (containing the bug) to those without FLake or with correct FLake as operationally only the parametrization with the bug was applied. The coming spring may show something because now the bug has been corrected while everything else remains unmodified in the operational HIRLAM system. Another problem is that there are not too much SYNOP stations making screen-level temperature observations in the immediate vicinity of the lakes so it is not easy to detect the impact in the verification statistics – these aspects where discussed by Eerola et al., 2014. Case studies might help, though. We mention this now shortly in the concluding section.

6. The use of in-situ observations is definitely of great value in the validation of lake surface state, however, when describing plans the authors might comment on the usability of satellite products as well. We added into the conclusions a sentence about the perspectives of using satellite products in the future.

Please also note the supplement to this comment:
https://www.geosci-model-dev-discuss.net/gmd-2018-270/gmd-2018-270-AC2-supplement.pdf

**Supplement:**

[revised manuscript text omitted]

---

## Editor Comment (EC1) · Jason Williams (Editor) · 21 Feb 2019

Dear Authors,

I have read through the referees reports and your responses and would like to draw your attention to an instruction in the requirements for referees and editors, namely:

" If an editor is presented with convincing evidence that the main substance or conclusions of a paper published in an editor's journal are erroneous, the editor should facilitate publication of an appropriate paper pointing out the error and, if possible, correcting it."

I am especially concerned about the general conclusions of referee 2 where he/she states: Due to this bug the model data related to ice behaviour and spring LSWT tem-

perature became unrealistic and therefore the corresponding results and discussions are of very limited interest.

As a journal we aim for papers of the highest quality therefore must apply some rigorous measures to protect the reputation of the journal. As it stands, the model data used for this paper has a significant drawback in that an error exists which directly affects the parameters under investigation. This significantly weakens any conclusions that can be taken away, resulting in a study which is not robust. For this reason the paper cannot be published in it's current form.

I would like to suggest two possible solutions as Topical Editor: (i) Use data from previous studies which can be directly linked e.g. Kheyrollah Pour et al. (2014) and/or Eerola et al. (2014).

(ii) You mention in your response that new results will be available in spring of this year. I suggest you use these simulations without the bug in the flake model and perform the validation exercise on this most recent model version to address the most major concern of the referees.

your sincerely,

Jason Williams.

---

## Author Comment (AC3) · 21 Mar 2019

In response to your
thank you for your comment and the possibility to improve our manuscript as well as to clarify our points of view in the open discussion. We appreciate your effort to maintain the high level of the journal. It is of crucial importance also for us, and the other authors from the lake modelling community, who initiated and contribute to this joint

special issue (GMD + HESS) on lakes in NWP and climate models.

However, we do not agree with your main conclusion: "As it stands, the model data used for this paper has a significant drawback in that an error exists which directly affects the parameters under investigation. This significantly weakens any conclusions that can be taken away, resulting in a study which is not robust." In fact, we did not agree either with the comment of the reviewer: "Due to this bug the model data related to ice behaviour and spring LSWT tem- perature became unrealistic and therefore the corresponding results and discussions are of very limited interest." Our mistake was not to reply clearly enough to this point. We will now explain viewpoint in detail.

The aim of our paper was to validate HIRLAM, which introduced FLake parametrizations in 2012. For this purpose, we gathered data from the archived at FMI operational HIRLAM output for more than six years 2012-2018 and compared them with a significant set of observations on lake surface state, as explained in the manuscript. Our results indicated, in particular, that according to the operational HIRLAM forecasts, the lake ice tended to melt too early in spring. We then started to analyse possible reasons for this. We found that there was no predicted snow on most of the lakes, in particular on those we selected for a deeper study and comparison with the ice thickess and snow depth observations. The next step was to try and understand why was this. As we happen to have access to the HIRLAM reference code and the code that was modified for the operational usage at FMI, we were able to see that in the code one value of a coefficient, that regulates the start of accumulation of snow on lake ice, was too large. In earlier experiments more than five years ago we had seen that the original value suggested by the FLake authors should be used instead of this value which was tried during preliminary testing.

Thus, the whole "bug" was this: instead of the original

h_Snow_min_flk = 1.0E-5_ireals , & ! Minimum snow thickness [m]

a test value value

h_Snow_min_flk = 1.0E-3_ireals , & ! Minimum snow thickness [m]

was by mistake left in the FMI operational version (and in the tagged version of HIRLAM v.7.4). This was corrected in the development code of the reference HIRLAM in 2014 but the correction never entered into the official v.7.4. It was perhaps our mistake to call this unsuccessful coefficient value a bug or technical error, which could give basis for a misunderstanding.

To summarize: we validated operational HIRLAM in extensive model-observation intercomparison. We found some results which did not correspond well to observations - the forecast lake ice tended to melt too early every year. In the HIRLAM data, we found almost no forecast snow on lake ice. We discussed the physics related to the role of snow on ice, which indicated that the missing snow may enhance melting of the ice in spring conditions. We were even able to suggest a probable reason for the missing snow, namely a too large value of a minimum snow thickness coefficient, which effectively prevented accumulation of snow on lake ice.

Now, after all this work, we were surprised do hear that the paper cannot be published because the results of HIRLAM using integrated FLake parametrizations did not correspond well the observations about melting of ice in spring! Our aim was to compare observations and model results, not to ensure that the (past) HIRLAM forecasts were ideal. As authors of a validation paper, we cannot undo six years of operational forecasts (which have by the way served well the weather service in Finland all these years) nor redo them. It is simply not our task. Once more: we have not run experiments, we have validated operational weather forecast model results on lakes. We have not run stand-alone FLake forced by HIRLAM output, but we have reported comparison against observations of the operational HIRLAM that contains integrated FLake parametrizations. None of the reviewers has suggested that our validation methods, or the way to extract model and observation data for the comparison were erroneus. In our opinion, our validation results are not of a limited interest. They give an important message to all developers of FLake and NWP models that snow on lake ice should be

treated carefully.

You suggest two solutions of the "problem": use data from earlier experiments or wait till summer 2019 to see if the HIRLAM forecasts improved after the correction of the coefficient value. Unfortunately, the HIRLAM experiment data used e.g. for the papers of 2014 are not available anymore. The published papers did not discuss the snow and ice depth in detail so it is not possible to refer to them more than we already did. We only found output files from one, unreported test experiment for January 2012 where within a month, maximum of 17 cm snow accumulated on ice of our lakes. Waiting till summer 2019 is not a good alternative for a couple of principal and practical reasons: 1) as stated earlier, we are not responsible for the FMI operational HIRLAM updates, and do not know if our suggested correction was made early enough to ensure improvements in this spring, 2) we cannot guarantee that this single correction will solve all problems of lake ice forecast in spring - based on earlier experiments we would say that the correction is a necessary but possibly not a sufficient condition for more exact lake ice break-up forecast, 3) waiting till the summer would mean that we might submit the corrected manuscript not before early autumn 2019, hoping that the correction entered the operational system early enough (if not, wait one more winter till summer 2020 ...).

In addition, we would like to remind that parametrization of the snow cover on lake and sea ice is perhaps the most complicated issue for the relatively simple ice schemes that are applied within the NWP models. In HIRLAM for example no snow parametrizations at all are applied over sea ice. Originally, also FLake was recommended to be applied on lake ice without the snow parametrizations. In our first experiments, reported in 2010, this was indeed the case, as it still is in some other applications of FLake in NWP models.

Now, in response to your concerns, we suggest this solution:

1) Through the manuscript, we corrected our unfortunate formulations related to "bug"

and "technical error" in order to avoid creating misunderstandings.

2) We wrote a short discussion section about snow on lake ice, with proper references with respect to the current results. We coordinated the conclusion section and the new discussion section to avoid overlap.

3) We checked the whole manuscript in order to make it crystal clear for the editors, reviewers and readers that we are validating operational model results, aiming at detecting problems and suggesting improvements for further developments.

At the same time, we took the opportunity to modify and add a few references, improve the terminology concerning the lake ice melting and freezing and make a few minor text corrections.

We would be grateful if you shared this reply with the three reviewers, too.

21.3.2019 Laura Rontu Kalle Eerola Matti Horttanainen

---

## Editor Comment (EC2) · Jason Williams (Editor) · 22 Mar 2019

Dear Authors,

Thank you for your response to my concerns regarding your manuscript which has been published. I should re-iterate that my main conclusion regarding the data used was based on the strong statements from the reports submitted from two out of the three independent referees (anonymous referees 1 and 2) who questioned why you were submitting a paper using results containing a significant error. The nature of the process is that the referees reports must be used by the editor to guide the review process. I am from the modelling community and a co-efficient which is not correct is commonly referred to as a bug.

[Figure]

Regarding my proposed solutions: (i) It is very unfortunate that the data is no longer available for you to perform an alternative analysis. However, in GMD the version of the model has to be very specific to avoid confusion regarding versioning of the code so possibly you would have been validating an old version no longer used by the rest of the community. (ii) You state that you are not responsible for the FMI code, but of course you could have offered co-authorship to someone who is responsible for such data in order to overcome this obstacle. (iii) In that the single correction possibly would not solve all the biases in the model is understandable, but you would be able to differentiate between the biases which exist due to other issues concerning the parameterization and that introduced by a co-efficient 100 times too high. (iv) The progress of each manuscript through the review process is not time limited. There are instances where the GMDD article does not proceed to GMD because requests from the referees and editors are not addressed.

I would also like to comment on some of the statements that the authors have made in their last paragraph of their reply (bottom of page C3). The fact that HIRLAM/Flake data does not capture observations well in the spring is not the reason that the major revision was requested, as I wrote in my initial comment posted in the discussion forum. It is that the data used for the validation contains an artifact that could present a misrepresentation of the biases in future forecasts for the performance around lakes. It is a completely valid reason to ask for further attention to the manuscript, especially considering the comments made by the referees and the admissions made by the authors that the impact of this error is not easily quantifiable. A validation paper serves as proof that when used in the future, there is confidence that HIRLAM can capture events to the first-order at least.

Finally, in light of the significant changes to the manuscript by the authors I am requesting additional input from the referees to ensure their major points have been adequately addressed.

Best regards, Jason Williams.

---

## Author Comment (AC4) · 20 May 2019

Please find updated replies and the traced differences between 1) the original and first revision, 2) the original and second revision and 3) the first and second revisions in the attached supplement file (pdf)

Please also note the supplement to this comment:
https://www.geosci-model-dev-discuss.net/gmd-2018-270/gmd-2018-270-AC4-supplement.pdf

---

## Author Comment (AC5) · 20 May 2019

Please find our updated replies to all three reviewers and the difference files in the reply to the first reviewer (AC4) and its supplement.

---

## Author Response (AR2)

[revised manuscript text omitted]

Second reply to the three reviewers

This manuscript has evolved from the original version of the 29th October 2018, which is available at the manuscript page, to the first revision of the 20th February 2019 and to the second revision of the 21st of March 2019.

Reviewers comments to the original revision were replied in detail on 20th February, available at the manuscript page. The revised manuscript is not available there.

Editor's comments on 21st February to the first revision were replied on 21 March 2019. The editor's comments, our reply and the editors reply are available at the manuscript page. A second revision was written in responce to the editors comments. It is not available at the manuscript page.

The essential differences between the first and second revisions were listed in the reply to the editor:

"1) Through the manuscript, we corrected our unfortunate formulations related to "bug" and "technical error" in order to avoid creating misunderstandings.  2) We wrote a short discussion section about snow on lake ice, with proper references with respect to the current results. We coordinated the conclusion section and the new discussion section to avoid overlap.  3) We checked the whole manuscript in order to make it crystal clear for the editors, reviewers and readers that we are validating operational model results, aiming at detecting problems and suggesting improvements for further developments.  At the same time, we took the opportunity to modify and add a few references, improve the terminology concerning the lake ice melting and freezing and make a few minor text corrections."

In our first reply, we did not react to the second reviewer's opinion that the results and discussions in this paper are of very limited interest due to a bug in treatment snow accumulation on ice. We presented our viewpoint on this in the reply to the editor on the 21st of March.

The terminology on lake ice melting and freezing dates was modified as originally suggested by the first reviewer, to melt-up and freeze-up, with a reference to Korhonen, 2019. In our first reply, we left this question open.

We introduced remarks related to the snow bug into several places of the first revision, as was also suggested by the reviewers. In the second revision, these are now mostly placed in a new discussion section about snow on ice instead of the scattered remarks.

About the technical side of FMI operational HIRLAM. Our suggested snow-on-ice correction was implemented into operations only 4.3.2019, not in October 2018 as we assumed. This means that it did not yet have time to influence properly in the results of lake ice melting this spring. At FMI, it has been decided that HIRLAM will be gradually decommissioned from operational usage during the next two years. This leaves the next winter 2019-2020 for the final proof of the influence of this modification in HIRLAM. However, we are convinced that the results of our unpublished experiment of January 2014, now mentioned in the new discussion section on snow on lake ice, did show that the modification works as expected.

To show the differences between the original, first and second revisions in a proper way, three difference pdf files were now generated by using latexdiff. They are: between the original and the

first revision, between the original and the second revision and between the first and the second revisions.

Please find below the original replies to three reviewers, with a few sentences in this color and highlighting to show the differences made between the first and second revisions in the context of earlier replies.

We hope these remarks and additional comparison documents make the situation clear. We apologize for not preparing and providing all needed files directly to the reviewers via the manuscript pages.

The 20th of May 2019

Laura Rontu, Kalle Eerola, Matti Horttanainen

Original replies
Reply to reviewer 1

Thank you for your careful reading of the manuscript, leading to helpful remarks and suggestions, which we mostly agree with. We have made modifications throught the whole text, but the kept the line numbers of the original manuscript in this reply. Please find our detailed response below. The difference between our original and corrected manuscript versions is provided in an attached diffpdf file.

General comments:
The paper presents results of HIRLAM (v7.4) model integrated to Flake model, lake surface state validation against in-situ observations of lake water temperature and ice cover during the period of 2012-2018 in Finland. In general, the paper structure is good and it is mainly written well. Same validation results against these in-situ observations have not been published earlier, eventhough some earlier papers have dealt with lake temperature and ice cover observation use in the HIRLAM. However, the noticed bug related to ice cover modelling is rather fundamental in physical way, and dominating the results, and makes me consider revising results with proper snowfall calculations on ice. It seems that in the future the HIRLAM model is no longer used and will be substituted with a new model. In that aspect, erroneous calculation could be documented in this article. The figures and tables could be improved and should be made more visual and reader-friendly; I will provide some specific comments on them. Especially figures and tables should run better in line with the text. Now, some figures are mentioned many pages before that they appear.

Concerning the snow-on-ice bug, it has now been corrected in the operational HIRLAM system, that continues running at FMI. The coming spring will provide material to check if the melting of lake ice is better handled. The operational correction was made on the 4th of March, 2019. thus this is not valid.

Also, in earlier experiments described e.g. in Kheyrollah Pour et al., 2014 and Eerola et al., 2014, this bug was not present. However, the results in those experiments were not validated against the ice and snow thickness, even the ice dates were used to a limited extent. In these circumstances, we do not consider it useful to run new HIRLAM experiments for checking the impact of the correction. Please note that in the new operational NWP at FMI, based on HARMONIE-AROME, no lake observations are analysed but Flake runs as it would in a climate model, i.e. continuing directly from the previous short forecast.

We will come back to the figures and tables when replying the specific comments. We agree that they should be improved. To correct the setup of figures at distant pages (caused by use of the default latex with template in the manuscript mode) we will ask advice from the GMD typesetting specialists if needed.

Specific comments:
1. Introduction, first paragraph (page 1, lines 16-19): Please provide some references.
We have first of all added references to papers describing the influence of lakes on weather forecasting in general, then influence on NWP and finally importance of describing the existence of ice correctly. We have selected the references so that they contain further relevant references.

2. You have used observations data for the year 2018 eventhough it is current year, usually provisional data. Is the in-situ data used in the analysis quality controlled? When the in-situ data was uploaded? And until which date the year 2018 data are used?
The operational analysis uses LWT observations from SYKE in real time. Those are quality controlled by the HIRLAM optimal analysis system: 1) excluding each station and comparing interpolated to its location nearby observations and 2) comparison against first guess. We use these quality-checked values from analysis feedback files in this study. Possible corrections by SYKE, made afterwards, were not used. The LID data and ice and snow thickness observations were obtained from SYKE open data base for this study, relying on their quality control:
LID was fetched 15.8.2018, snow depth 17.9.2018 and ice thickness 16.10.2018 from
http://rajapinnat.ymparisto.fi/api/Hydrologiarajapinta/1.0/
odataquerybuilder/

We added a sentence about the quality control and mention how the SYKE data was obtained.

3. Page 3, Figure 1. I would like to have it more visual-friendly. Is there certain meaning with arrow line thickness, if not then harmonize.

We now mention that the thin arrows are related to data flow between the HIRLAM analysis-forecast cycles while the thick ones describe processes within each cycle. We made also another correction to the Figure as suggested by reviewer 2.

4. Page 5, line 16. Please make reference to SYKE network, which year status it is?
(There are 34 sites in the network in year 2018 according to the SYKE database)
We explain this better in section 3.2.1. , i.e. that there are 34 stations now from which we use in the operational HIRLAM the original year 2011 selection that has never been changed since that. Originally, we excluded rivers and a couple of stations that then seemed to send data less regularly. The list needs to be updated for HARMONIE-AROME if LSWT analysis will be introduced there in the future.

5. Chapter 3.2.2. Freezing and melting dates. Article Korhonen (2006) has introduced terms for freezing and break-up in English, please use those. See: Korhonen, J. 2006. Long-term changes in lake ice cover in Finland. Nordic Hydrology 37(4-5): 347–363.
Thank you, we are aware of this terminology but selected freezing and melting according to the suggestion by our native linguistic advisor Emily Gleeson. In our earlier papers written together with our Canadian colleagues, we have used consistently the terms freeze-up and break-up. Now we did not like the suggested mixture of freezing and break-up, but perhaps there are good reasons to use this combination. We would like to leave the last word to our native British GMD editor of

the current manuscript Jason Williams. This is not valid, we now use freeze-up and break-up dates, following Korhonen, 2019.

6. Please state little bit more why these lakes were chosen. Were they only ones large enough to HIRLAM grid or were there other criteria?
The main criteria of selecting just these lakes for LID was the data availability: the most complete time series during the selected years, and a reasonable size that provided the best fraction of lake in HIRLAM grid. We now mention this.

7. I suggest combining figures 3 & 4 to same gridded figure with four graphs. Remove from temperature scale dots after the kelvin numbers. In figure caption open up meaning of fc an fob, an.
We kept two figures, that we consider to be more clear in the final two-column setup of the journal. The fc-ob-an were added to captions.

8. Chapter 4.2. is little bit hard to read/understand. Try to rewrite it more clear.
Thank you, we tried to clarify. This chapter is re-written totally to make it easier to read.

9. Page 10: Text paragraph, it is not clear what are different percentage categories.
Rewritten

10. Table 2: What are the units in this table?
Thank you, units added

11. I suggest combining Figures 5 and 6 to same figure (a and b)
Done

12. Page 12, last paragraph: make more clear in the text if you are talking about HIRLAM (analysed/forecast) or observed freezing and melting days.
Rewritten

13. Chapter 4.3. Make a reference to where lake area/depth records are taken. GLDB perhaps?
We renewed the list in Table A3 based on updated material for GLBD v.3 (not yet available at the Flake site but received by courtesy of Margarita Choulga), made the reference and mentioned it more clearly in the text.

14. Figures 7-10 could be combined to one gridded figure (a, b, c, d) Remove dots after Kelvin scale numbers.
We created 2 figures and removed the dots.

15. Figure 11. & 12. Add variable name and Unit in Y-scale. Just one legend could be below graphs since they are all same. For codes 28 and 29 use verbal definitions, please. It seems data until early 2018 is used?
Done. Data till summer is used 2018 (see above).

16. Figure 13. Add variable name and Unit in the Y-scale. In headings, use only lake name and years: Lappajärvi 2012-2013, Kilpisjärvi 2012-2013, Simpelejärvi 2012-2013.
Done

Technical/typo corrections:

1) Abstract: line 3 "integrated to HIRLAM" -> integrated into HIRLAM
Done

2) Use wording "in-situ" or "in situ" through whole text. Now there are both versions in the text.
Done

3) I would use "lake ice freezing and "lake ice melting" instead of lake freezing and melting (all text) (e.g. page 2, line 21)
Done

4) Page 2, line 31: I would consider revising wording "became available"
"were obtained"

5) Page 4, line 31: I would consider revising wording "basic material"
"is the basis of this study"

6) Figure 2. Page 6: Please note that abbreviation LID has not been introduced in the text before.
This a setup problem, now we repeat the definition in the caption, too.

7) Chapter 3.2.2 "codes 27-30" should not be used in the text or figures, use instead verbal definitions. Codes are so called database codes and not normally used as definitions. They are irrelevant as code numbers.
Replaced

8) Please check through the text that LWT and LSWT are used coherently. Page 13, line 1: LWST -> LSWT, Page 18, line 13 SYKE LSWT?

This is a bit problematic. Our idea was to call SYKE observations LWT because they are taken at the depth of 20 cm, not exactly at the surface that the satellite would have seen. However, Flake and HIRLAM analysis are dealing with LSWT even when the analysis is based on observed LWT. Perhaps the easiest solution is to call everything LSWT and mention the small difference when introducing the SYKE observations. We now did this.

9) Chapter 3.2.3 Ice thickness and snow depth on lakes
Done

10) Page 7, line 8: typo Simpelejärvi
Corrected

11) Chapter 3.3.1. Lake surface water temperature (LSWT)
Corrected

12) Page 8, line 2: Use verbal definition instead of category 29. Same in line 3 for category 28.
Done

13) Page 8, line 9: SYKE LWT observations
See 8

14) Page 8, line 21: typo known
Corrected

15) Page 15, line 9: 125 Wm-2 (superscript)
Corrected

16) Page 15, line 19: 2012-2018
? This is LaTex's work ...

17) Page 18, line 1: wrong -> incorrect/erroneous
Corrected

18) Page 18, line 17: ice thickness and snow depth
Corrected, also elsewhere

19) References: Please check that all references are formatted same way. For example, if many initial letters using space between or not in consistent way. I noticed some typos:
Thank you, corrected as suggested

Page 24,1. Potes, M. -> Potes, M.

Page 24, line 22. Gandin, L. missing :

Page 25, line 5. Remove ++ after Hydrology Research.

Page 25, line 11. co authors -> write all names

Page 25, line 33 et al > write all names

Page 25, line 33. Yang, Y., coauthors -> write all names and put the year in the end

Reply to reviewer 2

Thank you for your helpful remarks and suggestions, which we mostly agree with. We have made modifications throught the whole text, but the kept the line numbers of the original manuscript in this reply. Please find our detailed response below. The difference between our original and corrected manuscript versions is provided in an attached diffpdf file.

General comments:

Rontu et al. utilize archived forecast data (2012-2018) from the NWP model HIRLAM to validate the analysed and forecasted state of lakes with respect to observations

within a model domain. Due to unfortunate circumstances this specific HIRLAM version included a bug which prevented snow to accumulate on the lake ice. Due to this bug the model data related to ice behaviour and spring LSWT temperature became unrealistic and therefore the corresponding results and discussions are of very limited interest. Okay, it illustrates the importance of representing snow on ice when simulating lakes in cold climate conditions.

Indeed, this bug was not present in our earlier experiments, e.g. Eerola et al., 2014 nor is it there in the latest (development) version of HIRLAM reference code. Now it is corrected also in the FMI operational version, that will allow to check the situation during the coming spring. The operational correction was made on the 4th of March, 2019. thus this is not valid. We now discuss more in depth in the reply to the editor's comments why we disagree with your statement that "the corresponding results and discussions are of very limited interest". A new discussion section about snow on ice was added in the manuscript.

The manuscript is in general carefully written and can be considered as a useful guidance on how to validate the state of lakes in a NWP or climate model when corresponding in-situ observational data are available. The authors carefully describe uncertainties with respect to representativeness of observations and representation of lakes in a model domain. Also, they describe how ice conditions may be estimated based on other data. All this information can be valuable for scientists planning similar exercises for other combinations of model and lake observations.

Thank you for the positive comment, nice to hear that our methods are considered useful!

As the authors say it is a well known behaviour of FLake to overestimate summer LSWT. This is also seen in the presented results. However, it can not be excluded that part of those biases presented may be explained by for example any biases in near-surface temperature conditions in general. After all, the lakes represent only some 10% of the land area even in Finland so a bias in near-surface air temperature due to discrepancies in representation of land processes can also contribute to the presented biases. Thus, I would like to see a comment on the general near-surface temperature bias in this HIRLAM setup. The authors do comment on the quality of simulated down-welling short-wave irradiance but a comment on long-wave would also be relevant.

FLake works over the fraction of lake in each gridbox, driven by the average radiative and specific over-lake turbulent fluxes at the lake-atmosphere interface. The lake water and ice temperatures and other in-lake prognostic variables result from the Flake prognostic parametrizations. The resulting (diagnostic) LSWT represents the lake surface temperature in each gridbox, while the land-surface tile is taken care by other parametrizations (ISBA land-surface scheme), which also essentially solve the surface temperature from the equation of surface energy balance, taking into account also the heat conduction in ground. The grid-average screen-level temperature, that is regularly verified against observations, results from intelligent interpolation between the surface (e.g. LSWT) and the lowest model level temperature. In practice, the latter seems to dominate, but in principle, $T_{2m}$ could be wrong due to wrong LSWT but not vice versa. While there is no direct connection between the average (dominated by land surface) predicted surface temperature and LSWT, both might be inaccurate due to inaccurate atmospheric forcing. Wrong radiation transfer in the model, for example due to the cloudiness or incorrect handling of cloud-radiation interactions, biased near-surface air temperatures (at the lowest model level) or wrong turbulent fluxes in the atmospheric boundary layer could be sources of such inaccuracies.

Presumably, the shortwave radiation is the dominating factor for the lake water and ice thermodynamics during the year. In the equation of surface energy balance, the radiation fluxes are net fluxes over specific surface types, and these depend also on the prescribed surface properties, in our case e.g. on the lake ice and snow albedo. It would be worth while to perform sensitivity studies, e.g. with a single-column version of a NWP model, to reveal how Flake parametrizations would react to the inaccuracies of the atmospheric forcing and to quantify the related uncertaincies. This could be left for a further study for example in the framework of HARMONIE-AROME NWP system.

We added a sentence "Most importantly, FLake provides HIRLAM with the evolving lake surface (water, ice, snow) temperature, that influences the HIRLAM forecast of the grid-average near-surface temperatures." into the Flake description (Section 2.1). We also discuss the uncertainties related to atmospheric forcing where only the shortwave flux is now mentioned in the conclusions. We come back to the temperature aspect in the reply of your Kilpisjärvi comment.

Detailed comments:

Page 2, line 3: Sounds a bit strange to combine observed LSWT and simulated ice thickness to estimate fractional ice: " Fractional ice cover (lake ice concentration in each grid-square of the model) is estimated separately based on the analysed LSWT and the ice thickness predicted by Flake."
We improved our unfortunate formulation that allowed misunderstanding and relocated the explanations into their proper sections. There are two cases and two ways to estimate ice cover extent: in analysis, only LSWT exists, so it is used there in similar way that is done for SST – full ice concentration if the grid-average temperature is -0.5ºC, none when it is 0ºC and linearly in between. In the forecast by Flake, only ice thickeness is available. When it is larger than a small treshold value, the diagnostics decides that lakes existing in this gridbox are all frozen, i.e. the ice concentration is 1. There is a fraction of lakes in each gridbox, so the grid-scale ice fraction is obtained by multiplying the ice concentration with lake fraction. Thus, 'separately' meant: based on LSWT for analysis and based on ice thickness for forecast.

Page 5, line 15 19: Here you refer to Figure 2 for the first time but in the caption of Figure 2 you use the abbreviation LID which is defined later in the text. Please, e.g., introduce "lake ice dates" also in the figure caption for clarity.
Done

Page 8, lines 1-2: A bit strangely formulated sentence: "including in the comparison data over all months". Please make it more clear.
Done. The idea was that in the LSWT (obsa file) comparisons the winter months were excluded but here we used all data.

Pages 9-12, Section 4.2: The bug which prevents snow to accumulate on ice in this HIRLAM version will seriously affect all results presented in this section so I think it would be fair to the reader to comment on this in the beginning of this section although it has been mentioned in previous sections.
We now discuss the reasons for too early melting when showing the results here. This section has been largely rewritten. We have added a discussion section on snow on lake ice to explain this issue systematically.

Page 13, line 5: You say that "Lake Kilpisjärvi is an Arctic lake at the elevation of 473 m". This is a complex terrain area so the height difference between the real lake

and the model lake might contribute to estimated biases in temperature. What is the corresponding height of the HIRLAM grid box here? Would a height correction of temperature make any difference for the results?

The mean surface elevation of this gridbox where Lake Kilpisjärvi occupies around 40% of the area, is 614 m that is higher than the lake elevation because the lake is located in a valley surrounded by mountains. The diagnostic screen-level temperature, to which a heigh correction of temperature could be applied, plays no role in the model's air-surface energy exchange. To our understanding, there is no way in Flake to apply height corrections as part of the prognostic calculations or diagnosis of lake surface (snow, ice, water) temperature, also we are not aware of studies related to this issue.

The observed LSWT is evidently measured on the lake at the correct height. During the objective analysis, Kilpisjärvi LSWT is influenced by the observation on the lake and possibly on the nearby lakes, which are probably too far from here to really influence the analysis result. Differences in lake elevations could in principle be taken into account in the optimal interpolation, but this is not currently done. More detailed discussion about the objective analysis of LSWT can be found in the paper by Kheyrollah Pour et al. 2017.

We now mention the difference in Kilpisjärvi and grid-average elevations.

Figure 1: In the text it says that (page 2, line 33 – page 3, line 1) "the prognostic Flake variables are not corrected using the analysed LSWT, which would require advanced data assimilation methods" but in the figure it says "INDEPENDENT LAKE DATA ASSIMILATION IN AN INTEGRATED NWP + LAKE MODEL". I suggest to remove "DATA ASSIMILATION" here since that is not done according to the text. And ice cover is simply 0 or 1 when ice is present or not, right? So, this is not really a diagnostic estimation I would say. Or does this include something else which is not yet clear from the text. . .? Okay, becomes clear on page 4. Maybe better to refer to Figure 1 a bit later when the background to the figure is clear from the text.

We agree with the suggestion about "INDEPENDENT LAKE DATA ASSIMILATION" and replaced it with "OBJECTIVE ANALYSIS OF LSWT" in the figure. We also moved the figure and reference to it into Section 2.2.

Figure 11: Colour indications of freezing and melting dates in the caption (blue and red) do not fit with the figure (orange and magenta).

Corrected

Reply to reviewer 3

Thank you for your helpful remarks and suggestions, which we mostly agree with. We have made modifications throught the whole text, but the kept the line numbers of the original manuscript in this reply. Please find our detailed response below. The difference between our original and corrected manuscript versions is provided in an attached diffpdf file.

General comments:

The paper presents the detailed validation of the FLake model implemented in the HIRLAM NWP system, focusing mainly on the lake surface state and utilizing in situ measurements. The validation

period is considerably large spanning over six years and a large number of lakes are included in the investigation. The validation area covers only Finnish lakes, consequently results are referring to arctic conditions and might not be generalized to other climate regimes. The technical properties of the modelling system as well as the observational dataset are described properly. A lake water temperature assimilation scheme is also presented, however, it is mentioned that this is only a diagnostic product. Perhaps, the application areas of this product could be highlighted so that the purpose of it is clearer for the reader.

We added a sentence about the possible use of the diagnostic analysis into section 2.2.

During the validation, lake surface water temperature (LSWT), freezing and melting dates and ice thickness are investigated. Regarding LSWT results are in line with previous studies, namely an overestimation by FLake is pointed out. Freezing dates are simulated by an adequate precision, however, melting dates are poorly forecasted. The cause of this problem is enlightened during the investigation of the ice and snow thicknesses, namely due to a coding error snow is not accumulated on the ice surface. Physical consequences of this bug (missing insulation in winter and different albedo in spring) are well described.

Detailed comments:

1. Page 5 line 18: it is mentioned that water temperature is measured at 20 cm below water surface. Could the authors comment, whether this depth was used also in previous validation studies they are referring to (e.g. Kourzeneva 2014). Also, are there any difficulties in the validation when water is already frozen, but ice thickness is not reaching 20 cm?

Yes, we have always used the same SYKE observations in our papers. These observations are only available during the ice-free period as mentioned in Section 3.2.1. and were used only then. There may be gaps between the observed freezing and melting dates and the dates when LSWT observations are made. Also, the locations of LID observations and LSWT measurements are not necessarily the same at the lakes where both types of observation are available. We added a couple of sentences about this into Section 3.2.2

2. Page 10, line 8: "with an area of 1 km^-2" should be "with an area of 1 km^2"
Corrected

3. Page 13 line 14: "common to the majority of lakes" is a bit vague, "similar to the results averaged over all lakes" might more precise.
Corrected as suggested

4. Page 15, line 9: "125 Wm-2": "-2" should be superscript as one line above.
Corrected

5. Perhaps the authors could shortly comment, whether the bug revealed had any detectable impact on the forecasts of atmospheric variables (e.g. 2 m temperature) in the HIRLAM model in the six year period.

The problem is that we do not know, because there is no way to compare the results with Flake (containing the bug) to those without FLake or with correct FLake as operationally only the parametrization with the bug was applied. The coming spring may show something because now the bug has been corrected while everything else remains unmodified in the operational HIRLAM system. The operational correction was made on the 4th of March, 2019. thus this is not valid. Another problem is that there are not too much SYNOP stations making screen-level temperature observations in the immediate vicinity of the lakes so it is not easy to detect the impact in the verification statistics – these aspects where discussed by Eerola et al., 2014. Case studies might help, though. We mention this now shortly in the concluding section.

6. The use of in-situ observations is definitely of great value in the validation of lake surface state, however, when describing plans the authors might comment on the usability of satellite products as well.
We added into the conclusions a sentence about the perspectives of using satellite products in the future.

[revised manuscript text omitted]
 FLakemean water temperature K prog by FLakemixed layer temperature K prog by FLakebottom temperature K prog by FLaketemperature of upper layer sediments K prog by FLakemixed layer depth m prog by FLakethickness of upper layer sediments m prog by FLakethermocline shape factor - prog by FLakelake ice thickness m prog by FLakesnow depth on lake ice m prog by FLakeLSWT K diag by FLake= mixed layer temperature if no icelake surface~~

25  ~~temperature K diag by FLake uppermost temperature: LSWT or ice or snowLSWT K anal by HIRLAMflag value 272 K when there is icelake surface roughness m diag by HIRLAMscreen level temperature over lake m diag by HIRLAMscreen level abs.humidity over lake m diag by HIRLAManemometer level u-component over lake m diag by HIRLAManemometer level v-component over lake m diag by HIRLAMlatent heat flux over lake Wm$^{-2}$ diag by HIRLAMsensible heat flux over lake Wm$^{-2}$ diag by HIRLAMscalar momentum flux over lake Wm$^{-2}$ diag by HIRLAMSW net radiation over lake Wm$^{-2}$ diag by~~

30

35

[revised manuscript text omitted]
 FLakemean water temperature K prog by FLakemixed layer temperature K prog by FLakebottom temperature K prog by FLaketemperature of upper layer sediments K prog by FLakemixed layer depth m prog by FLakethickness of upper layer sediments m prog by FLakethermocline shape factor - prog by FLakelake ice thickness m prog by FLakesnow depth on lake ice m prog by FLakeLSWT K diag by FLake= mixed layer temperature if no icelake surface temperature K diag by FLake uppermost temperature: LSWT or ice or snowLSWT K anal by HIRLAMflag value 272 K when there is icelake surface roughness m diag by HIRLAMscreen level temperature over lake m diag by HIRLAMscreen level abs.humidity over lake m diag by HIRLAManemometer level u-component over lake m diag by HIRLAManemometer level v-component over lake m diag by HIRLAMlatent heat flux over lake Wm$^{-2}$ diag by HIRLAMsensible heat flux over lake Wm$^{-2}$ diag by HIRLAMscalar momentum flux over lake Wm$^{-2}$ diag by HIRLAMSW net radiation over lake Wm$^{-2}$ diag by HIRLAMLW net radiation over lake Wm$^{-2}$ diag by HIRLAMdepth of lake m pres in HIRLAM gridfraction of lake 0-1pres in HIRLAM gridfraction of lake ice 0-1diag in HIRLAM grid

[revised manuscript text omitted]

---

## Author Response (AR3)

Reply to the reviewers of the third round

Dear reviewers,

thank you for the positive and useful comments. We agree with them and have made corrections as detailed below. A few additional minor corrections were made as shown in the difference file between the third and fourth versions.

The manuscript untitled "Validation of lake surface state in the HIRLAM v.7.4 NWP model against in-situ measurements in Finland" from Rontu et al. is an interesting paper on a current topic: the representation of lakes in numerical weather prediction models. To my knowledge, it is the first time anyone has analyzed the results of more than 6 years of operational data of lake relevant meteorological parameters (Surface Temperature and freeze-up / break-up dates). It is an honest description of the results. I have carefully analyzed the authors' responses and the modifications made. In my opinion, the authors have fully answered the questions raised by previous reviewers. Therefore, I suggest that the article be published.

Anyway, I still suggest the following minor changes:

page 2 line 14: change 2016 for 2013.
O.K.

page 2 line 22: change "appears to be quite" for "is"
O.K.

page 3, line 16 change "near surface temperatures" for "near surface temperatures, humidity and wind"
O.K.

page 3, line 20 : "Bottom sediment calculations were excluded." is in contradiction with Apendix A. please explain or rectify

Modified Appendix A accordingly.

page 5, line 18: "After that the development of HIRLAM was frozen." please rewrite

Rewritten as: "After that further development of HIRLAM model has been stopped. "

page 6, line 4: change "on he" for "on the"
O.K.

page 9, line 1: "When subzero temperatures were excluded (...)" If the condition is this one, there are no reason to exclude many of the occurrences of 275 and 278 K…

Sorry, we did not get your point. Does the explanation of classes in Fig.3. help? According to observations and model, water temperatures between 0 and 5 oC are not uncommon in spring after break-up or before freeze-up in winter. The subzero water temperatures in the HIRLAM framework are used only as a flag of ice existence, not to denote real temperatures. Ahaa, perhaps you understood this as air temperature? Corrected to "When subzero LSWT values were excluded..."

page 9, label of figure 3: The classes are not well explained. It seems that there are 3 K intervals...  The same comment applies to the next figures.

Added into the caption: "classified in three-degree intervals from 270.1 to 303.1 K"

Page 9, line 3: change "situation" for "results" or …
"distribution"

Page 10, line 5: And what about the effect of a direct radiative heating of the bottom sediments?

Added "and possibly to the effect of the direct radiative heating of the bottom sediments"

Page 12, Table 3: The meaning opf the columns should be indicated in the label.

Added

Page 20, line 20: And what about the effect of a direct radiative heating of the bottom sediments?

Added: "In HIRLAM-FLake, the direct radiative heating of the bottom sediments is not taken into account, which may also contribute to this error."

Page 20, line 24: delete "evidently"
O.K.

Page 21, line 11: change " applied in the European" for " applied in some European"
O.K.

Page 21, line 13: Is HARMONIE the same as ALADIN_HIRLAM? Please explain what is HARMONIE?

Replaced with "This system uses the newest version of the global lake database (GLDB v.3)..." Otherwise, the story is complicated: there is an attempt to explain A-H system terminology in a paper in print for Adv. Sci. Res., 1, 1–8, 2019 https://doi.org/10.5194/asr-1-1-2019

[revised manuscript text omitted]